# Mind the Quote: Enabling Quotation-Aware Dialogue in LLMs via Plug-and-Play Modules

**Yueqi Zhang**[1]*, **Peiwen Yuan**[1]*, **Yiwei Li**[1], **Shaoxiong Feng**[2], **Xinglin Wang**[1], **Jiayi Shi**[1]
**Chuyi Tan**[1], **Boyuan Pan**[2]†, **Yao Hu**[2], **Kan Li**[1]†

[1]School of Computer Science and Technology, Beijing Institute of Technology
[2]Xiaohongshu Inc

{zhangyq,peiwenyuan,liyiwei,wangxinglin,shijiayi,tanchuyi,likan}@bit.edu.cn
{shaoxiongfeng2023}@gmail.com {panboyuan,xiahou}@xiaohongshu.com

## Abstract

Human–AI conversation frequently relies on quoting earlier text—"check it with the formula I just highlighted"—yet today's large language models (LLMs) lack an explicit mechanism for locating and exploiting such spans. We formalize the challenge as span-conditioned generation, decomposing each turn into the dialogue history, a set of token-offset quotation spans, and an intent utterance. Building on this abstraction, we introduce a quotation-centric data pipeline that automatically synthesizes task-specific dialogues, verifies answer correctness through multi-stage consistency checks, and yields both a heterogeneous training corpus and the first benchmark covering five representative scenarios. To meet the benchmark's zero-overhead and parameter-efficiency requirements, we propose QuAda, a lightweight training-based method that attaches two bottleneck projections to every attention head, dynamically amplifying or suppressing attention to quoted spans at inference time while leaving the prompt unchanged and updating $< 2.8\%$ of backbone weights. Experiments across models show that QuAda is suitable for all scenarios and generalizes to unseen topics, offering an effective, plug-and-play solution for quotation-aware dialogue.[1]

## 1 Introduction

Large language models (LLMs) have become powerful generalists [Marjanović et al., 2025, Chowdhery et al., 2023, Kamalloo et al., 2023], yet their behavior in *quotation-rich* conversation remains rudimentary [Lin and Lee, 2024]. In everyday conversation, users routinely refer back to earlier turns:

"That result looks wrong—check it with the **formula I just highlighted**."

Such requests simultaneously specify **where** the model should attend in the history and **how** the selected text should be used. Today's chat systems offer no principled mechanism for this: users must copy–paste snippets, insert disruptive markers, or hope the model guesses their intent.

We formalize the problem as **span-conditioned generation** (§2.1). Each turn is represented by the conversation history $H$, a set of quotation spans $\mathcal{R}$, and an intent utterance $U$. From this abstraction, we derive one **BASE** scenario and four orthogonal extensions—**MULTI-SPAN**, **EXCLUDE**, **INFO-COMBINE**, and **COREF**—that together cover real-world quoting behavior. Specific examples are shown in Fig. 1.

---

*Equal contribution.
†Corresponding author.
[1]Our code is available at https://github.com/marvelcell/MindtheQuote

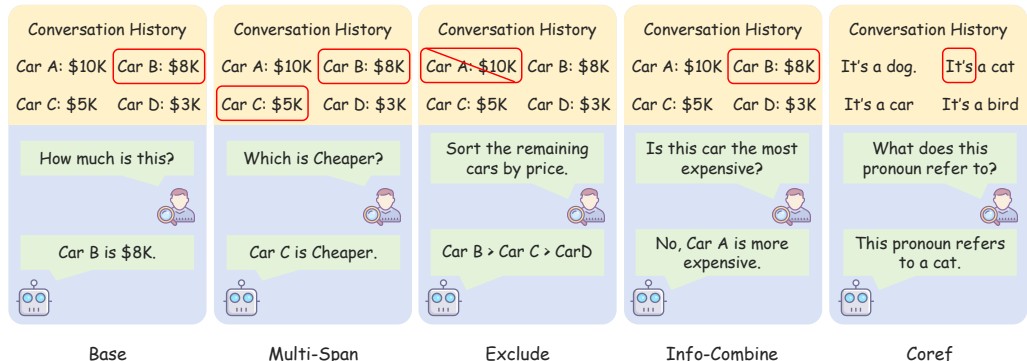

Figure 1: Illustrative examples of the five quotation scenarios.

Existing span-conditioned generation methods include CONCAT-REPEAT [Yeung, 2025], MARKER-INSERTION [Lin and Lee, 2024], and ATTENTION-STEERING (PASTA) [Zhang et al., 2023], all of which employ hand-crafted mechanisms (content repeat, boundary markers, or static attention re-weighting) without training. To evaluate their effectiveness, we construct a benchmark composed of five subsets using a carefully designed quotation-centric data pipeline. Given a topic and task specification, this pipeline synthesizes quadruples $\langle H, \mathcal{R}, U, \text{ANSWER} \rangle$ and verifies their internal consistency automatically, and ensures each instance is validated by human annotators. On this test bed, all three baselines succeed on only a subset of the tasks (The untrained part of Tables 4 and 5), highlighting the limitations of non-trainable methods in capturing in-span information and underscoring the need for a training-based, position-aware solution.

To fill this gap, we propose QUADA—a training-based method that embeds token-level span information directly into attention mechanism through inserted lightweight adapters. For every attention head, two bottleneck projections dynamically amplify or suppress scores adaptively on tokens that fall inside the quoted spans and inject a stronger retrieval signal for them. The original prompt remains untouched, and the inserted adapter adds fewer than 2.8% additional parameters to the backbone. Trained on a heterogeneous corpus generated by our pipeline, QUADA delivers state-of-the-art performance on all five quotation scenarios, generalizes to unseen topics, and attaches seamlessly to multiple model families.

Our contributions are summarized as follows:

1. We present the first position-aware formulation of quoting in conversation and conclude five diagnostic tasks that reflect nearly all practical use cases in the real world.
2. We propose an automated pipeline, equipped with built-in correctness checks, to generate a high-quality, human-verified benchmark and large-scale, fully-automated training corpora.
3. We propose a training-based, plug-and-play method that empowers LLMs to follow quoting intents without prompt inflation, delivering consistent gains across all scenarios.

## 2 Preliminary: The Formulation of Quotation-Guided Generation

### 2.1 Problem Setup

Let the tokenized conversation history be $H = (t_1, \ldots, t_n)$, where each $t$ represents a token. At turn $T$, the user supplies a set of **reference spans**:

$$\mathcal{R} = \{(s_l, e_l) \,|\, 1 \leq s_l \leq e_l \leq n, \ l \in \mathbb{N}\} \tag{1}$$

identified by token offsets in $H$, and a natural-language **intent utterance** $U$ that specifies the user's intent over those spans (e.g., summarize, compare, answer the question based on the spans). $s_l$ and $e_l$ in Eq. 1 represent the start and the end tokens' index of the $l_{th}$ quotation span.

Given the triplet $(H, \mathcal{R}, U)$, the model generates an answer $y = (y_1, \ldots)$ by maximizing

$$P_\theta(y \mid H, \mathcal{R}, U) \tag{2}$$

This formulation characterizes how the model generates responses in scenarios where specific spans from the conversation history are referenced and an intent is provided.

**Diagnostic sub-tasks**  To fully address the problem, we decompose quoting into a base scenario and four orthogonal extensions that together span practical quoting behaviors:

**Base**  The intent is associated with exactly one quoted span, representing the simplest scenario.

**Multi-Span**  The intent jointly involves multiple disjoint spans.

**Exclude**  The intent explicitly requests that a quoted span be ignored when answering.

**Info-Combine**  The intent asks the model to combine the quoted text with its related context, preventing over-focusing.

**Coref**  Resolution tasks where the quote might be a pronoun, similar word, or sentence whose meaning depends on its exact position.

The five diagnostic tasks cover four orthogonal axes of complexity that build on the Base primitive: (i) Multi-Span increases span cardinality; (ii) Exclude introduces negative selection; (iii) Info-Combine requires context-aware synthesis beyond the quoted text; and (iv) Coref demands token-level positional resolution. Taken together, they form a minimal yet complete basis that mirrors the ways users quote information in real conversation.

## 2.2  Existing Strategies for Span Injection

We analyze three existing strategies for injecting the triplet $(H, \mathcal{R}, U)$ into an LLM.

**(i) CONCAT-REPEAT**  For every $(s_l, e_l) \in \mathcal{R}$, verbatim content $H_{s_l:e_l}$ is prepended to $U$, yielding $U' = [H_{s_l:e_l}; U]$. Duplication conveys relevance but discards positional origin and inflates the prompt by $\sum_l (e_l - s_l + 1)$ tokens.

**(ii) MARKER-INSERTION**  Boundary tokens such as $\langle\texttt{EMPHASIZE}\rangle$ and $\langle\texttt{/EMPHASIZE}\rangle$ are inserted around each span. The modified history $H'$ is concatenated with $U$ and re-encoded. Position is explicit at the cost of roughly doubling tokens per conversation.

**(iii) ATTENTION-STEERING**  Pointer-Aware Sparse Tailoring keeps the textual prompt unchanged. Instead, it multiplies attention scores on a fixed subset of heads whenever a token's index falls inside any $(s_l, e_l)$. This produces an implicit positional bias with zero prompt overhead, but a single global scaling factor can limit the model's generalization capability in diverse scenarios and tasks.

We summarize the trade-offs of these methods in Table 1. An extended discussion of related works is provided in Appendix A.

Table 1: Qualitative comparison of span-injection baselines.

| Method | Prompt overhead | Injection mechanism |
|---|---|---|
| Concat-Repeat | high | content duplication (no position) |
| Marker-Insertion | high | explicit span markers |
| Attention-Steering | none | implicit head scaling |

# 3  Method

Our approach has two mutually–reinforcing components. First, a quotation-centric data pipeline generates large-scale training corpora and rigorously validates our benchmark that exercises the five scenarios introduced earlier. Second, a training-based position-aware module injects span information into a frozen LLM with minimal additional parameters.

## 3.1  Dataset Construction Pipeline

The pipeline turns a high-level specification into quadruples $\langle H, \mathcal{R}, U, \text{ANSWER}\rangle$ through five stages.

**Step 1: Attribute synthesis** To ensure that the data we generate is as generalizable and realistic as possible, following [Yu et al., 2023, Long et al., 2024], we prompt LLM to sample a structured attribute set that fully determines the forthcoming conversation and task. Table 2 lists the attributes and their corresponding value ranges. Detailed descriptions of specific attributes are provided in Appendix B. Each scenario is paired with its own task list, which is also generated by the LLM and stored with the attributes.

Table 2: Attributes for data generation.

| Category | Values / range |
|---|---|
| Topic | 102 topics (67 train / 35 test) |
| Tone | {neutral, informal, persuasive, ... } |
| History length | 1–10 turns |
| Information points | 2–10 per conversation |
| Task | {summarize, compare, rank, ... } |
| Span length | {word, sentence, paragraph} |

**Step 2: Conversation and Quotation span generation** Given the sampled attributes, the generator LLM (including o1, o1-mini [Jaech et al., 2024], o3-mini [OpenAI, 2025], GPT-4o [Hurst et al., 2024], Qwen-plus [Yang et al., 2024]) writes (i) a multi-turn conversation $H$ and (ii) an exhaustive list of token-offset spans covering every quotable information point, yielding the reference set $\mathcal{R}$. For the Coref scenario, we directly utilize the gold pronoun–antecedent spans from the CoNLL-2012 corpus [Pradhan et al., 2012], embedding them verbatim into $H$ to keep offsets valid.

**Step 3: Task-driven question and answer generation** Conditioned on $H$, $\mathcal{R}$, and the chosen TASK + SCENARIO, the LLM must (i) craft a single question $U$ applicable to all relevant span subsets, (ii) select span subsets that satisfy the scenario definition (1 span for BASE; $\geq 2$ for MULTI-SPAN, EXCLUDE, and INFO-COMBINE), and (iii) produce a correct answer for each subset. For Info-Combine, spans not selected serve as backgrounds.

**Step 4: Automatic validity checks** We cast $U$ into a multiple-choice format whose options are the answers from Step 3, then run three tests: (i) **Span sufficiency**—with only the chosen spans (plus context for Info-Combine scenario), the generator LLM must pick the correct option; (ii) **Span necessity**—without those spans, the LLM must fail or return multiple candidates; (iii) **Context requirement** (Info-Combine only)—a foil answer that ignores the related context must mislead the LLM when no background information is provided. A sample is accepted only if all tests pass.

**Step 5: Human verification** For the benchmark split, we add a manual audit layer. An LLM first rates every item against its attributes, flagging any item scoring below a threshold. Human annotators then inspect flagged cases, correcting or discarding them as needed. This extra pass ensures the benchmark's reliability while keeping annotation cost manageable.

Table 3: Number of samples per scenario.

| Scenario | Training set | | | Benchmark | | |
|---|---|---|---|---|---|---|
| | MCQ | Open-Ended | Total | MCQ | Open-Ended | Total |
| Base | 2200 | 2200 | 4400 | 500 | 500 | 1000 |
| Multi-Span | 2400 | 2400 | 4800 | 500 | 500 | 1000 |
| Exclude | 2400 | 2400 | 4800 | 500 | 500 | 1000 |
| Info-Combine | 2300 | 2300 | 4600 | 500 | 500 | 1000 |
| Coref | 2200 | 2200 | 4400 | 500 | 500 | 1000 |

The pipeline yields a heterogeneous training set and a human-validated benchmark, each containing multiple-choice and open-ended variants for the five scenarios. Table 3 reports the sample counts for each variant and scenario. By training on topics disjoint from the benchmark, we guarantee that models learn the *skill of quoting* rather than memorizing the content. To safeguard label correctness, the automatic validity checks described above remove 32.9%, 37.3%, 70.4%, 74.4%, and 15.2% of the initial candidates for the BASE, MULTI-SPAN, EXCLUDE, INFO-COMBINE, and COREF tasks,

respectively—an elimination rate we found essential for the reliability of the final corpus. Examples of each scenario are listed in Appendix D.

## 3.2 QUADA: Position-Aware Attention Modulation

Given our benchmark's fine-grained tasks, an effective model must satisfy three requirements: (i) introduce no prompt overhead, (ii) be parameter-efficient, and (iii) steer attention toward (or away from) quoted spans on demand. We satisfy these constraints with **Quotation Adapter (QUADA)**, a drop-in module that adds two lightweight bottleneck projections to every attention head (Fig. 2).

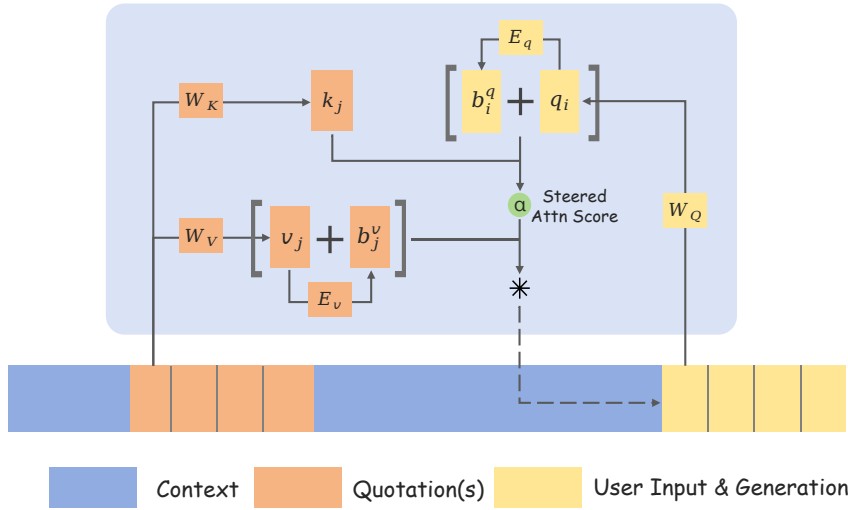

Figure 2: Overview of QUADA.

Let $q_i, k_j, v_j \in \mathbb{R}^d$ denote the query, key, and value vectors of a single attention head at positions $i$ and $j$ ($j \leq i$), and $d$ represents the dimensionality of the attention head. At inference time, compared with the non-trained methods, the model additionally receives the token-level span of the quotations and the user inputs:

$$\mathcal{R} = \{(s_l, e_l) \mid 1 \leq s_l \leq e_l \leq n, \ l \in \mathbb{N}\}, \quad and \quad Q = (s_q, \ldots), \tag{3}$$

where $\mathcal{R}$ is the set of quoted spans and $Q$ marks the span of the user's latest utterance, $s_q$ represents the start token's index of the question part. For notational convenience, we further define an indicator:

$$\mathbb{1}_{j \in \mathcal{R}} = \begin{cases} 1 & \text{if } j \in (s_l, e_l) \text{ for any } (s_l, e_l) \in \mathcal{R}, \\ 0 & \text{otherwise.} \end{cases} \tag{4}$$

**Query-side modulation** To enable the model to adjust the attention given to the quotation span adaptively, each head employs a bottleneck MLP $E_q: \mathbb{R}^d \to \mathbb{R}^d$ that yields a bias vector $b_i^q = E_q(q_i)$ for query vector. We modify only those queries originating from the current question and the generated tokens, i.e. $i \geq s_q$; otherwise $b_i^q = \mathbf{0}$. The resulting attention score is:

$$\left(q_i + \mathbb{1}_{j \in \mathcal{R}} \, b_i^q\right)^\top k_j = q_i^\top k_j + \mathbb{1}_{j \in \mathcal{R}} \, (b_i^q)^\top k_j, \tag{5}$$

The extra term $(b_i^q)^\top k_j$ selectively steers the attention score of quoted tokens, allowing the model to follow the user intent implicitly.

**Value-side enrichment** To strengthen the representation of quoted tokens, every head learns a second bottleneck MLP $E_v: \mathbb{R}^d \to \mathbb{R}^d$ that produces a strengthen vector $b_j^v = E_v(v_j)$ for the token belongs to the quotation spans. The value vector is then replaced by:

$$\tilde{v}_j = v_j + \mathbb{1}_{j \in \mathcal{R}} \, b_j^v. \tag{6}$$

The $\tilde{v}_j$ here can be computed directly from the cached $v_j$, no additional pass over the conversation history is required.

We only update these bottleneck MLP layers during training, which ensures that the inherent capabilities of the LLM remain unaffected. Despite its flexibility, the parameter of QUADA overhead is minimal. Let $r \ll d$ be the bottleneck width, each head then introduces $2dr + 2rd$ extra weights, keeping the total increase below $2.8\%$ for the 3B and $1.6\%$ for the 8B backbones used in our experiments.

# 4    Experiment

Table 4: Results of QUADA on Qwen2.5-3B-Instruct on all scenarios. Values in bold represent the best results, while values that are underlined represent the second-best results.

| | Methods | Base | | Multi-Span | | Exclude | | Info-Combine | | Coref | |
| | | Acc | Const | Acc | Const | Acc | Const | Acc | Const | Acc | Const |
|---|---|---|---|---|---|---|---|---|---|---|---|
| Untrained | VANILLA | 28.4% | 2.5 | 24.4% | 2.7 | 26.8% | 2.9 | 36.2% | 2.8 | 29.4% | 2.2 |
| | PASTA | 32.3% | 1.7 | 29.6% | 2.4 | 24.0% | 2.5 | 38.2% | 1.9 | 27.1% | 2.4 |
| | MARKER | 34.0% | 2.7 | 25.2% | 2.8 | 26.4% | 3.0 | 26.0% | 2.9 | 25.6% | 2.3 |
| | CONCAT | 83.2% | 4.3 | 67.8% | 4.2 | 15.2% | 2.3 | 38.2% | 3.0 | 30.4% | 2.3 |
| Training Based | PASTA | 34.0% | 2.0 | 29.7% | 2.6 | 22.4% | 2.9 | 39.0% | 3.1 | 32.7% | 2.5 |
| | MARKER | 44.8% | 4.1 | 26.8% | 3.3 | 7.8% | 2.6 | 23.0% | 2.7 | 53.4% | 3.2 |
| | CONCAT | 90.0% | **4.5** | 89.8% | 4.3 | 85.8% | 3.9 | 77.2% | 3.3 | 39.8% | 2.3 |
| | QUADA | **95.2%** | 4.4 | **94.6%** | 4.3 | **92.8%** | 4.2 | **85.8%** | 3.5 | **90.2%** | 4.1 |

Table 5: Results of QUADA on Llama3.1-8B-Instruct on all scenarios.

| | Methods | Base | | Multi-Span | | Exclude | | Info-Combine | | Coref | |
| | | Acc | Const | Acc | Const | Acc | Const | Acc | Const | Acc | Const |
|---|---|---|---|---|---|---|---|---|---|---|---|
| Untrained | VANILLA | 21.2% | 2.3 | 20.6% | 2.7 | 22.4% | 2.8 | 29.4% | 2.7 | 21.6% | 2.3 |
| | PASTA | 29.0% | 1.8 | 15.8% | 2.3 | 21.6% | 2.3 | 22.6% | 3.0 | 22.4% | 2.4 |
| | MARKER | 30.0% | 2.4 | 24.2% | 2.8 | 21.0% | 2.8 | 31.0% | 2.7 | 29.6% | 2.4 |
| | CONCAT | 78.2% | 4.2 | 68.6% | 4.2 | 25.0% | 2.3 | 34.4% | 3.2 | 31.8% | 2.8 |
| Training Based | PASTA | 27.1% | 2.4 | 25.5% | 2.9 | 25.0% | 3.2 | 31.2% | 3.1 | 27.5% | 2.5 |
| | MARKER | 82.8% | 4.4 | 83.6% | 4.3 | 63.4% | 3.9 | 56.0% | 3.5 | 81.6% | **4.0** |
| | CONCAT | 95.8% | 4.4 | 93.8% | 4.3 | 88.0% | 4.2 | 79.4% | 3.8 | 40.0% | 3.2 |
| | QUADA | **96.0%** | 4.5 | **98.2%** | 4.4 | **93.2%** | 4.4 | **82.6%** | 3.8 | **94.8%** | 3.8 |

## 4.1    Experimental Setup

**Backbone models**    We adopt two instruction-tuned LLMs with different scales and architectures: QWEN2.5-3B-INSTRUCT [Yang et al., 2024] and LLAMA-3.1-8B-INSTRUCT [Grattafiori et al., 2024]. For QUADA, we set the query and value side bottleneck width to $r = 256$, which introduces 75 M trainable parameters on Qwen ($2.8\%$ of the model) and 130 M on Llama ($1.6\%$). All backbone weights are frozen.

**Baselines and their trainable variants**    We benchmark QUADA against the three span-injection strategies from §2.2. Besides the original untrained versions, we devise trainable counterparts so that every method benefits equally from the synthetic corpus.

- **CONCAT-REPEAT.** We attach an auxiliary bottleneck to each attention head. If a token belongs to a duplicated span, its re-encoded representation is added to the main projection, mimicking explicit repetition while allowing gradients to flow.

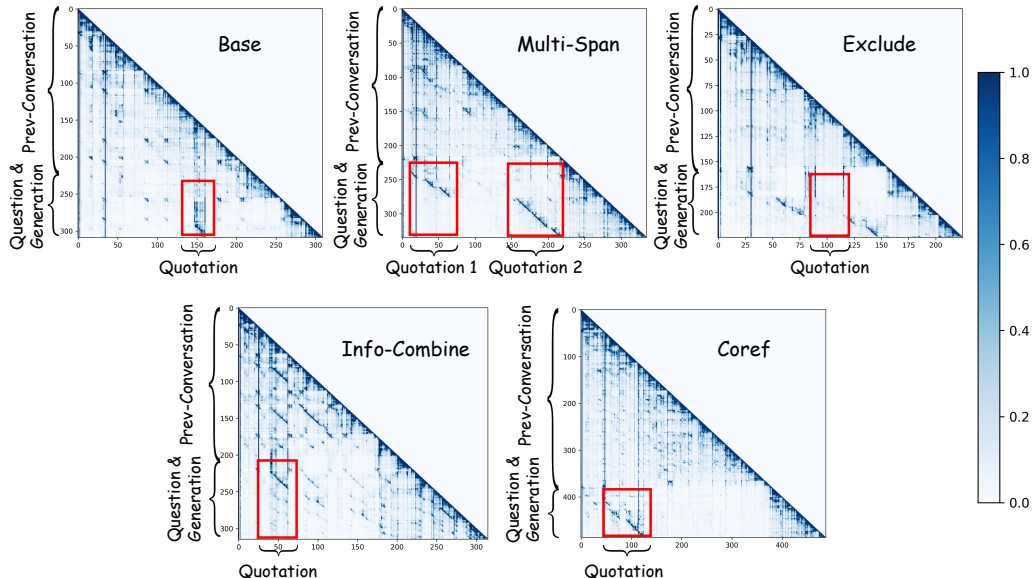

Figure 3: Averaged attention maps adjusted by QUADA. Attention scores are averaged and normalized over all attention heads. Red rectangles highlight the quotation span(s), while darker colors indicate stronger attention.

- **MARKER-INSERTION** The tokenizer is extended with two special symbols <emphasise> and </emphasise>. During fine-tuning, we update only their embeddings; all other parameters stay frozen, so the prompt overhead remains the same as in the untrained variant.
- **ATTENTION-STEERING (PASTA)** In the zero-shot setting, we reuse the 25 attention heads originally selected on the JSON FORMATTING data [Zhang et al., 2023]. For the trainable version, we rerank heads on our own training split and use the top 25 at inference time, following the protocol in the original method.

**Evaluation metrics** Each subset is paired with a metric that matches its output structure: We employ two metrics to assess these methods. *Accuracy (Acc)* for the single-choice questions. A prediction is correct if the chosen option exactly matches the gold. *Consistency score (Const)* is used for open-ended generations. Following recent best practice [Gu et al., 2024], we prompt a GPT-4o-mini model [Hurst et al., 2024] to rate the consistency between the model's output and the gold answer on a scale from 1 to 5. Higher scores indicate stronger alignment. The full judging prompt is provided in Appendix C.

Additional experimental details (including the GPU model, memory specifications, and criteria for human evaluators) are provided in Appendix F.

### 4.2 Main Results

**QUADA effectively handles diverse quoting behaviors** Our method consistently achieves the best performance across all five scenarios on both backbones (Tables 4 and 5). In **Base**, **Multi-Span**, and **Exclude** scenarios, QUADA accurately adapts its response behavior without requiring explicit rule-based control. In the **Info-Combine** scenario, where the model must integrate quoted spans with the related unquoted backgrounds, QUADA avoids over-focusing on the quote itself, demonstrating its capacity to use the quotations in a context-aware manner. Finally, in the **Coref** task, which requires sensitivity to the exact position of the quoted text, QUADA achieves 90.2% accuracy on Qwen and 94.8% on Llama, which are far ahead of all baselines, highlighting its ability to understand the position of the quotation span.

**Training is essential for the quotation task** Across all scenarios and both models, trained variants of all methods consistently outperform their untrained counterparts, demonstrating that span-conditioned behavior cannot be captured through static or untrained injection mechanisms alone. The superior performance of QUADA when trained on our synthetic corpus provides further validation for

the effectiveness of our training methodology and span-based attention modulation strategy, as well as confirms the quality and comprehensiveness of the training set.

**GPT-4o-mini judge is unbiased in our experiment**  To validate the accuracy of the 4o-mini-as-judge in our experiments, we randomly selected 25 data samples from each scenario and asked human evaluators to use the same rating criteria as the 4o-mini for evaluation. The results indicated a Pearson correlation coefficient of 0.75 between human and 4o-mini ratings, demonstrating that our 4o-mini evaluations for the Open-Ended task are generally unbiased relative to human judgment.

**QUADA preserves the model's generative fluency**  We additionally evaluated fluency with GPT-4o-mini. Both CONCAT-REPEAT and QUADA receive fluency ratings that are exceptionally clear, logically structured, and straightforward to read, indicating that our method does not degrade surface-level language quality.

**QUADA adapts to complex scenarios**  Building on the five basic scenarios above, we further evaluate three practical extensions: cross-lingual use, a multi-turn variant where successive questions each quote a different span, and mixed cases that combine MULTI-SPAN with EXCLUDE/INFO-COMBINE. In all cases, QUADA requires no task-specific rules or architectural changes and preserves its span-conditioned behavior. See Appendix E for details.

## 4.3  Analyses

**QUADA adaptively re-allocates attention**  Figure 3 plots the average head-wise attention for each diagnostic scenario. In **Base** and **Multi-Span**, the heatmap shows dark vertical bands exactly at the quoted offsets, while the rest of the context is uniformly suppressed—precisely what the tasks require. For **Exclude**, the pattern flips: the quoted segment turns pale, confirming that our method can down-weight forbidden spans. In **Info-Combine**, a hybrid pattern emerges: the quoted span still peaks, yet moderate attention is also maintained on other contexts, enabling the fusion of related information. Finally, in **Coref**, attention locks onto the quoted pronoun and its immediate neighborhood, providing the positional anchor needed for antecedent resolution. Together, these heatmaps show that QUADA dynamically adjusts attention according to the user's instruction. All visualized samples and their span annotations are provided in the Appendix D.2.

**Validity of the Auto-Generated Benchmark**  To assess how well our synthetic benchmark reflects real-world citation behavior, we manually curated a reference benchmark by filtering and adjusting a subset of samples for each scenario and question type. We then evaluated 10 diverse models (from Qwen, Llama, Phi families, etc.) on both the auto-generated and human-curated benchmarks, and computed the Pearson correlation between their scores.

As shown in Table 6, all correlation coefficients exceed $0.82$, with several above $0.95$, indicating a strong agreement between the automatically constructed and manually validated evaluations. These results confirm that our synthetic benchmark faithfully reproduces the relative difficulty and quotation behaviors seen in human-curated data, validating its suitability as an evaluation benchmark for real-world quotation scenarios.

Table 6: Pearson correlation between model scores on the LLM and human-based benchmarks.

|  | Base | Multi-Span | Exclude | Info-Combine | Coref | Avg. |
|---|---|---|---|---|---|---|
| MCQ | 0.83 | 0.95 | 0.97 | 0.93 | 0.87 | 0.91 |
| Open-Ended | 0.85 | 0.87 | 0.82 | 0.96 | 0.89 | 0.88 |

**QUADA generalizes across model scales**  Figure 4 reports the Accuracy and Consistency for Qwen2.5 series models from 1.5B to 14B parameters on all benchmarks. Even the 1.5B model surpasses 90% accuracy on three scenarios and averages $\sim 4.0$ consistency score. Performance rises steadily with scale, where the 14B variant achieves 98–99% on SINGLE and MULTI with a 4.5 consistency score. The monotonic trend indicates that QUADA delivers strong gains on small models and continues to improve as the model's parameter count grows.

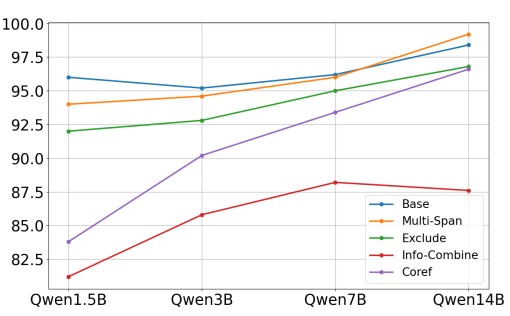 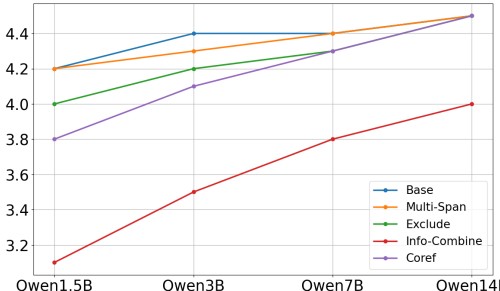

(a) Effect of model scale on multiple-choice accuracy.    (b) Effect of model scale on open-ended const scores.

Figure 4: Impact of backbone model size on QUADA performance across the Qwen 2.5 series models.

## 4.4 Ablation Study

**Query and Value Modulation are Complementary in QUADA**   We compare three variants of QUADA on Qwen2.5-3B-Instruct while keeping the total trainable parameters fixed in Table 7. The results show that **Value-only** modulation already recovers most of the full method's strength (e.g., 92.8% on MULTI and 87.6% on EXCLUDE), confirming that richer value vectors give quoted tokens a strong retrieval signal. **Query-only** modulation is much weaker, suggesting that steering attention scores in isolation cannot compensate for under-informative token representations.

The **full** method outperforms both ablations on every task, with the most significant margin on INFO-COMBINE (+10.8% over value-only). In this setting, the query branch first directs attention toward relevant regions; the value branch then supplies the content needed to merge quoted spans with surrounding context. We probe the representations by computing the **cosine similarity** between the bias $b^v$ and its base vector $v$ for every layer. The similarity averages **0.62** across layers, with **30%** of layers above **0.80**, and $\|b^v\| \approx 0.6\,\|v\|$. These statistics indicate that value modulation consistently reinforces quoted-span representations.

Table 7: Performance impact of query and value modulation components in QUADA.

| Variants | Base | Multi-Span | Exclude | Info-Combine | Coref |
|---|---|---|---|---|---|
| QuAda (query-only) | 78.8% / 4.3 | 58.4% / 3.7 | 49.6% / 3.8 | 44.6% / 3.0 | 69.4% / 3.7 |
| QuAda (value-only) | 94.8% / **4.4** | 92.8% / **4.3** | 87.6% / **4.1** | 75.0% / 3.4 | **91.6%** / **4.1** |
| QuAda (full) | **95.2%** / **4.4** | **94.6%** / **4.3** | **92.8%** / **4.2** | **85.8%** / 3.5 | 90.2% / **4.1** |

**QUADA is Parameter Efficient**   Here we examine how the size of the QUADA, controlled by the bottleneck dimension $r$, affects the performance of QUADA. Table 8 reports the results of varying the bottleneck width $r$ for the query and value projections in the Qwen2.5-3B-Instruct model. Notably, even at the smallest setting ($r$=64), QUADA already achieves over 94% accuracy on three out of five tasks and maintains a generation consistency score around 4.0. This indicates that most span-conditioning behavior can be captured with a highly compact QUADA module.

To balance accuracy and model size, we adopt $r$=256 as the default configuration: it consistently matches or sometimes outperforms all other settings across all tasks while remaining lightweight (< 2.8% of the backbone's parameters). These results demonstrate that QUADA can deliver substantial performance gains without a large parameter budget, making it especially suitable for resource-constrained deployments.

Table 8: Impact of bottleneck width $r$ on QUADA performance.

| QuAda ($r$) | Base | Multi-Span | Exclude | Info-Combine | Coref |
|---|---|---|---|---|---|
| $r = 64$ | **96.0%** / 4.4 | 94.2% / **4.4** | **95.0%** / **4.2** | 84.4% / 3.4 | 87.8% / **4.1** |
| $r = 128$ | 94.8% / **4.5** | 94.2% / **4.4** | 93.6% / **4.2** | 80.4% / 3.5 | 90.0% / **4.1** |
| $r = 256$ | 95.2% / 4.4 | 94.6% / 4.3 | 92.8% / **4.2** | **85.8%** / 3.5 | 90.2% / **4.1** |
| $r = 512$ | 95.4% / 4.4 | **94.8%** / 4.3 | 93.8% / **4.2** | 84.8% / 3.5 | **90.6%** / **4.1** |

# 5 Conclusion & Future Work

We introduced **span-conditioned generation** as a principled formulation for quotation-rich conversation and released a diagnostic benchmark covering five orthogonal quoting behaviors. To facilitate research in this new setting, we developed an automatic data-generation pipeline capable of synthesizing large-scale, high-quality training corpora and human-validated benchmarks. Building on this data, we proposed QUADA, a lightweight adapter that injects positional signals via query and value side adapters without inflating prompts or substantially enlarging the backbone. Extensive experiments across two instruction-tuned LLMs and multiple model sizes (from 1.5B to 14B) demonstrated that QUADA consistently outperforms both training-free and training-based baselines, maintains fluency, and scales gracefully. Taken together, our results suggest that this work can substantially advance human–LLM interaction, without introducing adverse side effects. Potential backdoor and misinformation risks, along with our mitigation measures, are discussed in Appendix G.

The present work is confined to monomodal, monolingual quotation. Our adapter is trained and evaluated solely on English text spans; we have not experimented with other modalities (images, tables, code) or multilingual settings—such as quoting a French sentence within an English dialogue. Extending QUADA to multimodal and cross-lingual quotation will require new benchmarks and may reveal additional challenges for position-aware attention, which we plan to explore in future work.

### Acknowledgments

This work is supported by Beijing Natural Science Foundation (No.4222037, L181010).

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

# A Related Works

## A.1 Attention-Guided Generation in Decoder-Only LLMs

A growing body of mechanistic-interpretability work shows that individual self-attention heads specialize in distinct generative roles[Zheng et al., 2024]. For instance, "retrieval heads" that gather long-range evidence [Wu et al., 2024] or heads whose ablation removes knowledge conflicts [Jin et al., 2024]. Such span-sensitive behavior motivates our adapter.

The most closely related control method is PASTA [Zhang et al., 2023, 2024], which boosts a fixed set of heads whenever their keys fall inside a user-marked span. *Focus Directions* [Zhu et al., 2025] extend this idea by adding a learned vector to key/query activations, steering attention toward salient tokens without prompt markup. Both approaches inspire our query-side biasing, yet neither enriches value vectors nor handles multi-span or negative constraints, gaps that QUADA fills.

**Latent-space steering**

A complementary line of research controls decoder-only LLMs by editing internal activations. Liu et al. [2023] recasts in-context learning examples as a single in-context vector that is added to hidden states, achieving stronger task transfer with zero prompt overhead. More generally, Subramani et al. [2022] extracted steering vectors from pretrained LMs can nearly perfectly reconstruct target sentences, and enable sentiment transfer via simple vector arithmetic. Extending this idea, Turner et al. [2023b] formalises activation engineering and introduces ACTIVATION ADDITION, a lightweight method that shifts activations along a "love → hate" direction to control sentiment without optimisation. Konen et al. [2024] demonstrates analogous style vectors that steer output toward specific writing styles. Beyond style, Li et al. [2023] identifies a "truthfulness" direction: nudging activations along it doubles LLaMA accuracy on TruthfulQA at inference time. Latent editing has also been used to modify factual knowledge: Hernandez et al. [2023] learn encodings of new facts and injects them into hidden layers so that generation aligns with the edited knowledge. Finally, Turner et al. [2023a] provides a systematic recipe for computing activation directions that steer toxicity, refusal, or politeness.

These works share our goal of parameter-efficient, inference-time control; however, they target global attributes (style, truthfulness, knowledge) rather than **span-conditioned quoting**. QUADA differs by selectively modulating query and value activations according to user-specified spans, enabling fine-grained, multi-span, and negative constraints that latent steering alone cannot express.

# B Attributes for Data Generation

**Topics**

```
Algebra
Calculus
Probability and Statistics
Mechanics
Electromagnetism
Optics
Thermodynamics & Statistical Mechanics
Inorganic Chemistry
Organic Chemistry
Biochemistry
Environmental Chemistry
Pharmaceutical Chemistry
Constitutional Law
Civil Law
Criminal Law
Administrative Law
Business Law / Commercial Law
International Law
Intellectual Property Law
Labor Law
Environmental Law
```

```
Arbitration and Mediation
Mechanical Engineering
Electrical Engineering
Electronic Engineering
Chemical Engineering
Civil Engineering
Materials Engineering
Aerospace Engineering
Biomedical Engineering
Computer Engineering
Environmental Engineering
Microeconomics
Macroeconomics
Econometrics
International Economics
Development Economics
Financial Economics
Labor Economics
Behavioral Economics
Public Economics
Environmental & Resource Economics
Nutrition and Diet
Sports Medicine & Fitness
Epidemiology
Public Health
Mental Health
Health Education
Health Management
Basic Clinical Medicine
Geriatrics & Rehabilitation
Health Policy
General Psychology
Developmental Psychology
Social Psychology
Clinical & Counseling Psychology
Cognitive Psychology
Personality Psychology
Physiological Psychology
Industrial/Organizational Psychology
Educational Psychology
Health Psychology
Marketing
Finance
Accounting
Management
Operations Management
Human Resource Management
Business Strategy
Entrepreneurship & Innovation
International Business
E-commerce
Cell Biology
Molecular Biology
Genetics
Physiology
Ecology
Developmental Biology
Microbiology
Zoology
Botany
Neurobiology
Ancient Philosophy
Modern Philosophy
Analytic Philosophy
Continental Philosophy
```

```
Ethics
Metaphysics
Epistemology
Political Philosophy
Philosophy of Science
Aesthetics
Data Structures & Algorithms
Operating Systems
Computer Networks
Software Engineering
Database Systems
Artificial Intelligence
Machine Learning
Computer Graphics
Distributed Systems & Cloud Computing
Cybersecurity
```

**Tone**

```
Formal
Casual
Persuasive
Analytical
Creative
Narrative
Enthusiastic
```

**Tasks**

```
##### Task for Base Scenario #####
Specified Passage Summarization: Summarize the single quoted passage
    into a concise statement. The user wants a brief overview or main
    idea derived exclusively from the quoted text
Specified Segment Q&A: Answer a question based only on the single
    quoted text. The user's query should be resolved strictly using
    information found in that quoted segment
Specified Segment Definition Extraction: Identify and restate the
    definition of a term or concept mentioned in the quoted text. The
    user is asking for a precise definition contained within that
    segment
Keyword or Key-Phrase Extraction: Extract the most relevant keywords
    or key phrases from the single quoted segment. This task focuses
    on highlighting crucial points or topics directly mentioned
Quoted Segment Rewriting: Rewrite the quoted segment while retaining
    its essential meaning. The user ask for a more readable or
    differently styled version of the same text
Quoted Passage Simplification: Simplify the quoted passage while
    retaining its essential meaning. The user may ask for a more
    concise, more readable, or differently styled version of the same
    text
Quoted Segment Sentiment Analysis: Analyze the emotions or attitudes
    expressed in the quoted segment. Identify and describe the
    feelings, such as happiness, sadness, anger, or neutrality,
    conveyed in the quoted region
Quoted Segment Data Extraction: Extract requested data or details from
     the quoted segment. Focus on retrieving specific facts or figures
     mentioned in the text
Quoted Segment True-or-False Verification: Present a factual statement
     about the single quoted text, and ask the model to determine
    whether it is True or False based strictly on that text. This can
    be verified by checking if the statement directly aligns or
    contradicts with the content in the quoted passage.
Quoted Passage Step-by-Step Procedure Extraction: If the single quoted
     text describes a procedure or a set of ordered steps, request the
```

model to list each step or stage in the correct sequence. The
correctness can be checked by confirming each listed step appears
(in the right order) within the quoted region.
Specified Segment Contradiction Detection: Present a statement or
claim and ask whether it contradicts the single quoted text, is
supported by it, or is not mentioned at all. This is easy to
verify: simply compare the statement to the quoted text to see if
it's in direct conflict, alignment, or absent.
Specified Information Existence Detection: Present a statement or
claim and ask whether it appears in the quoted text. In this task,
the statement refers to information that exists in the unquoted
portion of the original context but not in the quoted segment. By
comparing the statement to the quoted text, the model should
determine if the statement is indeed absent there.
Quoted Passage Named Entity Identification: Ask the model to extract
and list all named entities (e.g., people, locations,
organizations) exactly as they appear in the single quoted text.
The correctness is judged by directly checking the text to see if
any named entities are missed or incorrectly added.
Quoted Segment Key Fact Listing: Instruct the model to list out the
key factual points explicitly mentioned in the single quoted text
(e.g., important dates, statistics, proper names, etc.). The user
then checks if each item in the model's list appears verbatim (or
unambiguously) in the passage, ensuring no extra or missing facts.

##### Task for Multi-Span Scenario #####
Multi-Source Information Comparison: Compare or contrast two or more
separate quoted passages (e.g., comparing their data, opinions, or
attributes). The user's question focuses on differences or
similarities across these quoted segments
Multi-Source Consolidated Summarization: Provide a unified summary
that covers all key points from two or more quoted passages. The
user wants an integrated overview of multiple fragments
Temporal (Time-Order) Reasoning: Determine or explain the
chronological order of events or facts mentioned across multiple
quoted segments. The user's question involves identifying which
event or statement occurred first or last
Contrasting Viewpoints Analysis: Identify differing viewpoints or
stances across multiple quoted regions. The user wants to see how
authors or speakers in those passages disagree or differ in
perspective
Merged Key Point Listing: Extract the core points from each quoted
passage and then compile them into a single combined list. The
user wants a concise set of bullet points that captures all cited
texts
Numerical/Data Comparison Across Sources: Identify and compare
numerical values or data points mentioned in two or more quoted
passages. The user's question focuses on determining the highest,
lowest, or most significant value among these sources.
Numerical/Data Sorting: Extract numerical values or data points from
two or more quoted passages and arrange them in a specified order
(e.g., from high to low or low to high). The user's question
focuses on presenting the data in a sorted list.
Multi-Source Condition Fulfillment: The user poses a set of conditions
(e.g., "Condition A is stated in quotation 1, Condition B is in
quotation 2"). Ask the model to determine if those conditions are
satisfied or met collectively, based strictly on the referenced
segments.
Multi-Passage/Sequence Reconstruction: If multiple quoted segments
each describe different parts or phases of a single process or
timeline, the user asks to arrange these parts in a coherent
sequence or flow. Correctness can be verified by checking if each
step appears in the right chronological or logical order as
indicated by the individual passages.

```
##### Task for Exclude Scenario #####
Summarize After Ignoring Selected Passages: The user designates one or
    more passages to be "ignored." The task is to summarize only the
    remaining, non-ignored content. The answer must exclude or skip
    any information from the ignored portions.
Sensitive Information Hiding: The user selects private or confidential
    text or daat to be hidden. The final output must not reveal the
    ignored data. The user wants only the non-sensitive parts to be
    disclosed.
Partial Anonymization or Redaction: The user specifically wants
    certain fields or details (like names, addresses, IDs) to be
    redacted. The task is to remove or mask those sensitive elements
    while preserving the rest of the information.
Summarize Unignored Sections: The user designates one or more passages
    to be ignored. The model must produce a concise summary only of
    the content not in those ignored passages. The ignored content
    should not influence or appear in the summary.
Sort or Rank Unignored Data/Number: After ignoring specific segments,
    the user asks the model to sort or rank remaining items based on a
     certain criterion (e.g., numerical values, alphabetical order,
    etc.). Only the leftover textual details (not ignored) should be
    used to perform the sorting or ranking.
Extract Keywords from Non-Ignored Content: The user designates some
    passages to be ignored and then requests keyword extraction. The
    model must parse only the remaining (unignored) text to identify
    relevant or significant keywords without referencing any ignored
    sections.
Named Entity or Concept Extraction (Non-Ignored Only): The user
    designates specific passages to ignore. The model must identify
    named entities (people, locations, organizations, etc.) or key
    concepts only from the remaining text. Any details found in
    ignored content should be excluded.
Non-Ignored Outline Generation: The user instructs the model to create
     an outline (e.g., bullet points or a structured plan) of the
    leftover information after certain portions are ignored. The
    resulting outline must reflect only the visible (unignored)
    details.
Compare Remaining data/number After Ignoring: The user designates
    certain passages to be omitted. The question is then to compare
    the data/number in only the remaining information. The solution
    should not incorporate any data from the ignored content.

##### Task for Info-Combine Scenario #####
Reference Justification Extraction: Given a reference quotation that
    contains a recommendation or statement (e.g., "Then you should
    rest early") and a background quotation that provides the
    underlying context (e.g., "I am very tired from work today"), the
    model must extract and clearly justify the reasoning behind the
    reference. The answer should explain that the suggestion is made
    because of the background's stated condition.
Price Comparison Based on Quoted Reference: When the reference
    quotation specifies a particular item (e.g., "Item B: 8 dollar")
    and background quotations list prices of other items (e.g., "Item
    A: 10 dollar" and "Item C: 5 dollar"), the model must compare
    these figures. The answer should identify which background item is
     more expensive than the referenced one, strictly based on the
    provided numerical data.
Temporal Sequence Reasoning with Context: With a reference quotation
    indicating an event with a specific date (e.g., "Event Y occurred
    on March 5th") and background quotations listing other events with
     their dates (e.g., "Event X occurred on March 1st" and "Event Z
    occurred on February 28th"), the model must determine the
    chronological relationship. The answer should identify which event
     happened before or after the reference event, solely using the
    provided dates.
```

Attribute Comparison Across Context and Reference: When the reference
    quotation provides a measurable attribute of an item (e.g., "
    Product Y has a rating of 4.0 stars") and background quotations
    offer similar attributes for other items (e.g., "Product X: 4.5
    stars" and "Product Z: 4.8 stars"), the model must perform a
    direct comparison. The answer should specify which products have
    higher (or lower) values than the referenced product.
Contextual Cause-Effect Linkage: Given a reference quotation that
    states an effect or recommendation (e.g., "You should replace the
    cooling fan immediately") and a background quotation that provides
     a cause (e.g., "The device overheated due to prolonged usage"),
    the model must link the effect to its cause. The answer should
    clearly state that the recommendation is based on the background
    cause.
Contextual Selection Based on Criteria: Provided a reference quotation
     that highlights a particular preference or criterion (e.g.,
    selecting an item with a specific price point) along with
    background quotations listing various items and their attributes,
    the model must select the items from the background that meet the
    criterion specified by the reference. The answer should list only
    those items that conform to the given requirement.
Direct Comparison with Context Integration: With a reference quotation
     that specifies a particular detail (e.g., "Item B costs 7.3
    dollar") and background quotations that offer comparable details
    for other items (e.g., "Item A costs 20 dollar" and "Item C costs
    2 dollar"), the model must directly compare these details. The
    answer should indicate which background item stands out relative
    to the reference.
Integrated Inference from Reference and Context: The model receives a
    reference quotation that implies a decision or inference (e.g., "
    You should leave earlier tomorrow") alongside background
    quotations that supply supporting context (e.g., "Traffic is
    usually heavy during rush hour"). The answer should integrate both
     to deduce the rationale behind the reference statement.
Contextual Elaboration of a Reference Statement: Given a reference
    quotation that provides a brief statement (e.g., "It would be wise
     to update the software") and a background quotation that offers
    additional detail (e.g., "The system has been experiencing bugs
    due to outdated software"), the model must elaborate on the
    reference. The answer should explain that the recommendation is
    based on the issues mentioned in the background.
Quantitative Comparison with Contextual Data: When a reference
    quotation contains a quantitative figure (e.g., "Service B costs
    20 dollars") and background quotations list similar figures for
    other services (e.g., "Service A costs 25 dollars" and "Service C
    costs 15 dollars"), the model must perform a quantitative
    comparison. The answer should specify which background figure
    exceeds or falls below the reference figure.
Relative Ranking Determination: With a reference quotation providing a
     metric for an item (e.g., "Model Y has a battery life of 8 hours
    ") and background quotations offering comparable metrics for other
     models (e.g., "Model X: 10 hours" and "Model Z: 7 hours"), the
    model must rank the items. The answer should clearly state which
    models rank higher or lower than the referenced one based on the
    provided metrics.
Conditional Outcome Reasoning: Provided a reference quotation that
    states a conditional suggestion (e.g., "If you experience
    overheating, turn off the device immediately") along with
    background quotations that supply the condition (e.g., "The device
    's temperature has been rising steadily"), the model must deduce
    the correct outcome. The answer should confirm the conditional
    recommendation based solely on the provided background.
Multiple Criteria Filtering: When a reference quotation emphasizes a
    specific criterion (e.g., "I prefer an affordable product") and
    background quotations list several items with various attributes (

```
                e.g., price, quality), the model must filter the background to
        identify only those items meeting the reference criterion. The
        answer should list the items that match the affordability
        requirement as provided in the background.
Cross-Domain Integration: The model is given a reference quotation
        from one domain (e.g., "You should upgrade your hardware")
        alongside background quotations from a related domain that offer
        supporting technical details (e.g., "The current system has a
        processing speed of 1.2 GHz"). The answer should integrate both
        sets of information to justify the recommendation, strictly
        relying on the provided details.
Contradictory Evidence Resolution: Given a reference quotation
        presenting a definitive statement (e.g., "The meeting is scheduled
         for 3 PM") and background quotations that offer conflicting
        information (e.g., "The organizer mentioned a delay, setting the
        meeting at 4 PM"), the model must resolve the contradiction. The
        answer should clarify the correct detail by logically prioritizing
         the background evidence.
Reference-Context Synthesis for Decision Making: In this task, the
        reference quotation poses a decision prompt (e.g., "Should I
        choose Option B?") and background quotations offer supporting
        details (e.g., pros and cons or specifications of Options A and B)
        . The model must synthesize the information from both to recommend
         a decision. The answer should clearly state which option is
        preferable based solely on the integrated information from the
        reference and background quotations.
```

## C   Full Prompt for Consistency Evaluation

**Prompt for Evaluating Base, Multi-Span, and Exclude Scenarios**

```
You are a fair evaluator. You will be given two pieces of text: Model
        Answer and Ground Truth. Your task is to assess how accurately the
         Model Answer reflects the essential facts and conclusions of the
        Ground Truth. Use the following five-level scale to guide your
        decision, then provide your final evaluation as a single line in
        the format "Score X" with no additional explanation or content.

Score 1: The answer is overwhelmingly incorrect or contradicts the
        Ground Truth in most respects, showing fundamental
        misunderstandings or factual errors, so that little to no
        essential information aligns with the Ground Truth.
Score 2: The answer contains multiple significant inaccuracies or
        contradictions, offering only a limited amount of correct
        information, and overall demonstrates unreliable alignment with
        the Ground Truth despite some partial correctness.
Score 3: The answer demonstrates moderate alignment with the Ground
        Truth, correctly capturing certain important points while also
        containing noticeable errors or omissions, so that its overall
        accuracy is only partially reliable.
Score 4: The answer is largely accurate with only minor mistakes or
        overlooked details, consistently reflecting the Ground Truth's
        main facts, thereby showing a good degree of correctness and
        trustworthiness.
Score 5: Answer used the same evidence as the Ground Truth and reached
         a conclusion identical to the Ground Truth.

After reviewing the Model Answer and comparing it to the Ground Truth,
         choose the best matching level from 1 to 5. Output your final
        rating as:
Score X
where X is the chosen number, and do not add any other text, marker or
         explanation in your response.
```

```
Ground Truth:
{gt_label}

Model Answer:
{predict}
```

**Prompt for Evaluating Info-Combine Scenarios**

```
You are a fair evaluator. You will be given two pieces of text: Model
    Answer and Ground Truth. Your task is to assess how accurately the
     Model Answer reflects the essential facts and conclusions of the
    Ground Truth. Use the following five-level scale to guide your
    decision, then provide your final evaluation as a single line in
    the format "Score X" with no additional explanation or content.

Score 1: The Model Answer reaches a different conclusion from the
    Ground Truth, and the referenced information is entirely unrelated
    .
Score 2: The Model Answer reaches a different conclusion from the
    Ground Truth, and some of the reference information is the same as
     that in the Ground Truth.
Score 3: The Model Answer reaches a different conclusion from the
    Ground Truth, and most of the reference information is the same as
     that in the Ground Truth.
Score 4: The Model Answer reaches the same conclusion as the Ground
    Truth, but it does not utilize the same information as the Ground
    Truth.
Score 5: The Model Answer reaches the same conclusion as the Ground
    Truth and utilizes the exact same information.

After reviewing the Model Answer and comparing it to the Ground Truth,
     choose the best matching level from 1 to 5. Output your final
    rating as:
Score X
where X is the chosen number, and do not add any other text, marker or
     explanation in your response.

Ground Truth:
{gt_label}

Model Answer:
{predict}
```

**Prompt for Evaluating Coref Scenarios**

```
You are tasked with evaluating the quality of an Answer provided for a
     coreference resolution problem. The evaluation should be based on
     how well the Answer aligns with the Ground Truth (correct answer)
    . Use the following scoring rubric to assign a score from 1 to 5:
Score 1: The Answer is completely incorrect. It claims that the
    pronoun does not refer to anything, or the response is entirely
    irrelevant.
Score 2: The answer points to an incorrect referent, which is not the
    one specified in the Ground Truth.
Score 3: The answer identifies several potential referents, including
    the one specified in the Ground Truth, along with additional
    irrelevant ones.
Score 4: The answer correctly identifies the referent specified in the
     Ground Truth, but contains inaccuracies or omissions in
    describing what it did or what happened.
Score 5: The answer correctly identifies the referent as defined in
    the Ground Truth and accurately describes what it did or what
    happened.
```

```
After reviewing the Model Answer and comparing it to the Ground Truth,
    choose the best matching level from 1 to 5. Output your final
    rating as:
Score X
where X is the chosen number, and do not add any other text, marker or
    explanation in your response.

Ground Truth:
{gt_label}

Model Answer:
{predict}
```

# D  Examples for the input samples

## D.1  Multi-Choice Question Examples

**Base**

```
{
    "instruction": "I've been reading about how social contracts
        influence public welfare policies. Could you explain where
        this idea originated?<|im_end|>\n<|im_start|>assistant\
        nCertainly! The idea of the social contract originates from
        philosophers like Thomas Hobbes, who argued that individuals
        agree to surrender some freedoms in exchange for security
        provided by the state. In 'Leviathan,' Hobbes described life
        without such an agreement as 'solitary, poor, nasty, brutish,
        and short.' This foundational principle laid the groundwork
        for understanding the relationship between citizens and their
        governments.
    <|im_end|>\n<|im_start|>user\nThat's fascinating. Can you give me
        an example of how this might work in practice today?<|im_end
        |>\n<|im_start|>assistant\nSure! Imagine a country called
        Altruria, which recently implemented a universal basic income
        program. The rationale behind this decision was rooted in
        social contract theory-the government reasoned that providing
        financial stability would reduce societal unrest and foster
        trust among citizens. By ensuring everyone had a safety net,
        Altruria aimed to fulfill its part of the implicit agreement
        between rulers and ruled.<|im_end|>\n<|im_start|>user\nBut isn
        't there always someone who disagrees with these kinds of
        agreements? How do we address those concerns?<|im_end|>\n<|
        im_start|>assistant\nAbsolutely valid point. Not all citizens
        may see eye-to-eye with the assumptions underlying social
        contract theory. For instance, critics argue that enforcing
        collective agreements can undermine individual autonomy. They
        believe people should have the right to opt out if they
        disagree with the terms imposed by the majority. This tension
        highlights the ongoing challenge of balancing communal
        obligations with personal freedoms within any society governed
         by social contracts.<|im_end|>\n
    <|im_start|>user\nBased solely on the information provided in the
        quotation. What does this passage suggest about the social
        contract?\nA. In the context of Altruria, the social contract
        involves the government providing financial stability through
        programs like universal basic income to reduce societal unrest
         and foster trust among citizens. It reflects an implicit
        agreement where the state fulfills its duty by ensuring a
        safety net for everyone.\nB. The social contract originates
        from philosophical ideas where individuals surrender some
        freedoms to the state in exchange for security, as argued by
```

```
            Thomas Hobbes. Life without such an agreement is described as
            harsh and unstable, emphasizing the importance of this
            arrangement between citizens and governments.\nC. Critics of
            the social contract argue that enforcing collective agreements
             can limit individual autonomy, suggesting people should have
            the right to opt out if they disagree with majority-imposed
            terms. This highlights the tension between communal
            obligations and personal freedoms within societies governed by
             social contracts.\nD. The quoted segments does not address or
             is insufficient to answer this question.\nPlease analyze the
            question first, then provide the answer in the format: 'The
            answer is X.'",
    "input": "",
    "output": "Analysis of the question:  \n- The passage specifically
            discusses Altruria implementing a universal basic income
            program as a means of addressing societal unrest and fostering
             trust, framed in the context of social contract theory.  \n-
            Option A directly aligns with the given passage, as it
            accurately describes the social contract in Altruria focusing
            on government responsibility to provide financial stability
            and a safety net in exchange for societal trust.  \n- Option B
            , while it contains ideas related to the broader concept of
            social contract theory (such as Hobbes's views), includes
            information not mentioned or implied in the passage. This
            makes Option B incorrect when judged strictly on the passage.
             \n- Option C introduces a critical perspective on the social
            contract, implying tension between collective agreements and
            personal freedoms. However, no such criticism or alternative
            perspective is mentioned or implied in the given passage. This
             makes Option C incorrect as well.  \n\nThe correct choice
            must focus solely on the details of Altruria and the specific
            interpretation of the social contract provided in the passage.
             Since the quoted segment provides enough information to
            answer this question, option D is not selected.  \n\nThe
            answer is A.",
    "quote_token_span": [
        [
            139,
            208
        ]
    ],
    "question_token_span": [
        321,
        535
    ]
}
```

**Multi-Span**

```
{
    "instruction": "How do serotonin and dopamine influence emotional
        regulation differently? Can you provide some insights into
        their roles and impacts?<|im_end|>\n<|im_start|>assistant\
        nCertainly! Let's delve into how serotonin and dopamine play
        distinct roles in emotional regulation. Firstly, serotonin
        plays a crucial role in stabilizing mood and promoting
        feelings of well-being and happiness. It helps regulate
        anxiety, sleep, and appetite. Low levels of serotonin are
        often associated with depression and anxiety disorders.
        Secondly, dopamine is involved in reward-motivated behavior
        and pleasure centers in the brain. It influences motivation,
        attention, and movement. High levels of dopamine can lead to
        increased energy and focus but can also contribute to
        impulsive behavior. When serotonin levels are low, individuals
         may experience heightened anxiety, irritability, and
```

difficulty sleeping. This can significantly affect daily functioning and overall quality of life. Conversely, elevated dopamine levels can result in enhanced motivation and alertness. However, excessive dopamine activity has been linked to conditions like schizophrenia and addiction. Therapeutically, selective serotonin reuptake inhibitors (SSRIs) are commonly used to treat depression by increasing serotonin availability. For conditions involving dopamine dysregulation, medications that modulate dopamine receptors are often prescribed.<|im_end|>\n<|im_start|>user\nUsing only the details given in the quotations, describe the primary effects of the neurotransmitter discussed and any associated risks or benefits mentioned.\nA. The neurotransmitter discussed is involved in reward-motivated behavior and pleasure centers in the brain.

Elevated levels of this neurotransmitter can result in enhanced motivation and alertness but also pose risks such as impulsive behavior, schizophrenia, and addiction.\nB. The neurotransmitter discussed plays a crucial role in stabilizing mood and promoting well-being. Therapeutically, selective serotonin reuptake inhibitors (SSRIs) are commonly used to treat depression by increasing the availability of this neurotransmitter, addressing issues related to low levels.\nC. The neurotransmitter discussed influences motivation, attention, and movement. For conditions involving dysregulation of this neurotransmitter, medications that modulate dopamine receptors are often prescribed to manage symptoms like excessive energy and focus, which can also contribute to impulsive behavior and conditions like schizophrenia and addiction.\nD. The quoted segments does not address or is insufficient to answer this question.\nPlease analyze the question first, then provide the answer in the format: 'The answer is X.'",
"input": "",
"output": "The analysis of the question involves carefully examining the quotations provided:\n\n1. The first quotation states that dopamine is involved in reward-motivated behavior, attention, motivation, movement, and pleasure centers. It specifically adds that high levels of dopamine can lead to increased energy and focus, but also impulsive behavior. No mention is made of schizophrenia or addiction in the quotations.\n   \n2. The second quotation refers exclusively to serotonin and discusses its therapeutic modulation through selective serotonin reuptake inhibitors (SSRIs) for treating depression. It does not mention dopamine or any of its associated effects.\n\nNow, let's compare the options:\n- **Option A**: This describes dopamine in terms of reward-motivated behavior and its potential effects, which partially match the quotations. However, it incorrectly adds conditions such as schizophrenia and addiction, which were not mentioned in the quotations. This makes Option A incorrect.\n\n- **Option B**: This describes serotonin and stabilizing mood, which matches the second quotation but refers to serotonin rather than dopamine. However, the question revolves around the neurotransmitter in the first quotation (dopamine), making Option B irrelevant.\n\n- **Option C**: This accurately matches the first quotation by highlighting dopamine's influence on motivation, attention, movement, and its association with symptoms like excessive energy, focus, and impulsive behavior. It also notes the use of medications to modulate dopamine receptors for conditions involving dysregulation of dopamine. This aligns completely with the given information, but it does not mention unrelated conditions like schizophrenia or addiction.\n\n- **Option D**: The quoted segments are indeed sufficient to answer this

```
        question, and since one of the other options (C) is correct,
        Option D is not selected.\n\nThe answer is C.",
    "quote_token_span": [
        [
            89,
            135
        ],
        [
            196,
            239
        ]
    ],
    "question_token_span": [
        245,
        470
    ]
}
```

**Exclude**

```
{
    "instruction": "In the context of Physiological Psychology, could
        you provide details on the levels of serotonin, dopamine, and
        norepinephrine in various brain regions and discuss their
        impact on cognitive functions? Please ensure that each
        neurotransmitter's discussion includes its specific brain
        region and the cognitive function it influences.<|im_end|>\n<|
        im_start|>assistant\nCertainly. Let us examine the levels of
        these neurotransmitters and their impacts: Serotonin levels in
         the hippocampus play a crucial role in memory formation.
        Higher levels of serotonin have been associated with improved
        long-term memory retention and consolidation. Dopamine levels
        in the prefrontal cortex are essential for decision-making
        processes. Elevated dopamine activity enhances cognitive
        flexibility and facilitates better problem-solving skills.
        Norepinephrine levels in the amygdala significantly influence
        emotional regulation. Increased norepinephrine can heighten
        alertness and responsiveness to emotionally charged stimuli,
        impacting how we process and react to emotions. Each
        neurotransmitter's presence in these specific brain regions
        contributes uniquely to cognitive functions.<|im_end|>\n<|
        im_start|>user\nIgnoring the selected parts, which
        neurotransmitter and brain region play a crucial role in the
        specified cognitive function?\nA. Serotonin levels in the
        hippocampus play a crucial role in memory formation. Higher
        levels of serotonin have been associated with improved long-
        term memory retention and consolidation.\nB. Dopamine levels
        in the prefrontal cortex are essential for decision-making
        processes. Elevated dopamine activity enhances cognitive
        flexibility and facilitates better problem-solving skills.\nC.
         Norepinephrine levels in the amygdala significantly influence
         emotional regulation. Increased norepinephrine can heighten
        alertness and responsiveness to emotionally charged stimuli,
        impacting how we process and react to emotions.\nD. The
        remaining information does not address or is insufficient to
        answer this question.\nPlease analyze the question first, then
         provide the answer in the format: 'The answer is X.'",
    "input": "",
    "output": "Analysis of the question:  \nThe question asks which
        neurotransmitter and brain region play a crucial role in the
        specified cognitive function. Based on the provided
        information:  \n- Dopamine levels in the prefrontal cortex are
         explicitly mentioned as being essential for decision-making
        processes, enhancing cognitive flexibility and problem-solving
         skills.  \n- There is no mention of serotonin levels in the
```

```
                    hippocampus or their role in memory formation in the provided
                    conversation.  \n- There is no mention of norepinephrine
                    levels in the amygdala or their influence on emotional
                    regulation in the provided conversation.  \n\nThe answer is B
                    .",
        "quote_token_span": [
            [
                    81,
                    111
            ],
            [
                    141,
                    183
            ]
        ],
        "question_token_span": [
                205,
                370
        ]
}
```

**Info-Combine**

```
{
        "instruction": "Could you provide a detailed breakdown of the
                current pricing for key diagnostic instruments used in
                Physiological Psychology research?<|im_end|>\n<|im_start|>
                assistant\nCertainly. To start, the EEG system costs $20
                ,000.<|im_end|>\n<|im_start|>assistant\nIn addition, the fMRI
                session is priced at $1,200.<|im_end|>\n<|im_start|>assistant\
                nMoreover, the PET scan usage fee is set at $3,000.<|im_end|>\
                n<|im_start|>assistant\nThe GSR equipment, which is essential
                for monitoring skin conductance, is available for $500.<|
                im_end|>\n<|im_start|>assistant\nFurther, the Heart Rate
                Monitor is offered at $300, providing a cost-effective option
                for cardiovascular assessments.<|im_end|>\n<|im_start|>
                assistant\nFinally, the Eye Tracking System stands at $800,
                rounding out the suite of diagnostic instruments.<|im_end|>\n
                <|im_start|>user\nWhich diagnostic instrument(s) cost more
                than the one(s) mentioned in the selected quotations? Build
                your answer solely from the data presented in the selected
                quotations and the related information.\nA. You referenced the
                 EEG system, which costs $20,000, and the fMRI session, priced
                 at $1,200. Comparing these with the background data, no other
                 instrument exceeds the cost of the EEG system. However, the
                PET scan, priced at $3,000, is more expensive than the fMRI
                session. Therefore, the PET scan is the only additional
                instrument more expensive than the one you have referenced.\nB
                . You referenced the GSR equipment, costing $500, and the
                Heart Rate Monitor, priced at $300. From the background data,
                all other instruments-EEG ($20,000), fMRI ($1,200), PET scan (
                $3,000), and Eye Tracking System ($800)-are more expensive
                than either the GSR equipment or the Heart Rate Monitor. Thus,
                 the EEG system, fMRI session, PET scan, and Eye Tracking
                System are all more costly than the ones you referenced.\nC.
                Based on the information provided in the quotations, no
                diagnostic instruments costing more than the ones mentioned
                are specified. The GSR equipment costs $500, and the Heart
                Rate Monitor costs $300. Therefore, there is no mention of any
                 instrument costing more than these within the given
                quotations.\n\nAnswer: None.\nD. More than one option is
                correct, or all are wrong. \nPlease output the analysis of
                each option first, then choose the correct option in the
                following format: 'The answer is X.'(Do not include other
                marker)",
```

```
    "input": "",
    "output": "**Analysis of Option A:**  \nOption A references the
        EEG system ($20,000) and the fMRI session ($1,200). However,
        the quotations specifically mention the GSR equipment ($500)
        and the Heart Rate Monitor ($300), not the EEG or fMRI.
        Therefore, this option does not correctly address the issue
        raised in the quotations, as it focuses on instruments not
        mentioned in the quotations. This makes **Option A incorrect
        **.\n\n**Analysis of Option B:**  \nOption B correctly
        identifies that the GSR equipment ($500) and the Heart Rate
        Monitor ($300) are referenced in the quotations. It then
        compares these prices to the background data, noting that all
        other instruments (EEG at $20,000, fMRI at $1,200, PET scan at
         $3,000, and Eye Tracking System at $800) cost more than
        either the GSR equipment or the Heart Rate Monitor. This
        analysis aligns with the provided information and effectively
        addresses the question. Thus, **Option B is correct**.\n\n**
        Analysis of Option C:**  \nOption C claims that no diagnostic
        instruments costing more than the ones mentioned in the
        quotations are specified. This contradicts the background data
        , which clearly shows that all other instruments (EEG, fMRI,
        PET scan, and Eye Tracking System) are more expensive than the
         GSR equipment ($500) or the Heart Rate Monitor ($300).
        Therefore, **Option C is incorrect**.\n\n**Analysis of Option
        D:**  \nOption D suggests that the quoted segment is
        irrelevant to the question, so it cannot be answered. However,
         since the quoted information directly relates to the question
         by referencing specific costs of GSR equipment and Heart Rate
         Monitors, **we do not choose Option D**.\n\nThe answer is B
        .",
    "quote_token_span": [
        [
            95,
            116
        ],
        [
            122,
            144
        ]
    ],
    "question_token_span": [
        176,
        534
    ]
}
```

**Coref**

```
{
    "instruction": "What is the world economic outlook for 1998
        according to Kazuo Nakamura's analysis?<|im_end|>\n<|im_start
        |>assistant\nXinhua News Agency, Beijing, January 16, by wire
        -LRB- reporter Wenxin Fan -RRB- Kazuo Nakamura, Director of
        Research at Japan's Dai - Ichi Kangyo Bank, said here today
        that the real growth rate of the world economy in 1998 will be
         about 3 %. Kazuo Nakamura, at a public lecture jointly
        organized by Beijing's Wu Jia Economic Consulting Company, Ltd
        . and Japan's Dai - Ichi Kangyo Bank, made a report concerning
         the new year's world economic and financial situation. After
        he reviewed the world economic and financial situation of '97,
         he said that due to the economic growth in Asia, including
        Japan, further slowing down, the economic growth of America,
        even though its current economic activity is relatively good,
        and of other countries will also feel a very large effect.
        Compared with the rapid growth of the past, the world economy
```

```
            has entered a period of adjustment. He forecast that the real
            growth rate for the world economy this year would fall from
            the 3.9 % of '97 down to 3 %. Nakamura thought that the rate
            of economic growth of Asia's countries and regions, with Korea
             and Thailand being the most prominent, would slide downward
            to a large extent. But, he said, for these countries the
            currencies already have devalued to a great degree, exports
            have recovered, and imports could decrease. Because of this,
            he expected that their economies would slip downward to the
            lowest point in the next half year. He estimated that the rate
             of economic growth in ASEAN would be about 4.5 percent.
            Nakamura said that the Japanese economy exhibited negative
            growth in the year '97, and that in '98, if it didn't adopt
            good economic policies, the economy would continue to be in
            negative growth. He estimated it would be negative 0.5 percent
            , plus or minus. He said further that China's economic growth
            would also slow in speed, feeling the effects of the economies
             of Japan and Asia being depressed and of the devaluation of
            these countries' currencies. But because China's usual income
            and expenditures show a favorable balance, and in addition it
            has over 100 billion US dollars in foreign exchange reserves,
            it isn't likely to sink into the crisis like the other Asian
            nations. -LRB- End -RRB-<|im_end|>\n<|im_start|>user\nWhat
            does the pronoun I selected refer to in the conversation?\nA:
            negative growth\nB: China's\nC: Japan's\nD: Can not be sure.\
            nPlease output the analysis of each option first, then choose
            the correct option in the following format: 'The answer is X
            .'(Do not include other marker)",
    "input": "",
    "output": "The pronoun refers to \"Japan's.\" In the context, the
            sentence discusses the Japanese economy's performance in 1997
            and its prediction for 1998. The statement specifies that \"if
             it didn't adopt good economic policies, the economy would
            continue to be in negative growth.\" Here, \"it\" logically
            refers to \"Japan's economy,\" as the preceding discussion is
            about Japan's economic state and the conditional requirement
            of adopting good policies to avoid further negative growth.
            The other options do not align with the subject of the
            sentence or the logical flow of the discussion.\n\nThe answer
            is C.",
    "quote_token_span": [
        [
            386,
            386
        ]
    ],
    "question_token_span": [
        517,
        583
    ]
}
```

## D.2 Open-Ended Question Examples

**Base**

```
{
    "instruction": "I've been reading about memory management in
            operating systems. Can you explain what \"page stealing\" is?
            It sounds intriguing but also a bit aggressive!<|im_end|>\n<|
            im_start|>assistant\nAh, \"page stealing\" is actually quite
            clever despite its name. When an operating system runs out of
            physical memory, it may need to reclaim pages from processes
            that aren't actively using them. Instead of waiting for a
```

```
        process to explicitly release memory, the system preemptively
        takes-or \"steals\"-unused pages and assigns them to other
        processes in need. This helps maintain performance without
        forcing applications to constantly monitor their own memory
        usage. Quite ingenious, isn't it?<|im_end|>\n<|im_start|>user\
        nThat makes sense! Now, I heard about this theoretical OS
        called ZyloSys. Does it do anything special with memory
        management?<|im_end|>\n<|im_start|>assistant\nIndeed, ZyloSys
        takes things further by implementing a hybrid approach. It
        combines traditional paging with selective swapping to
        optimize both speed and resource utilization. Specifically,
        ZyloSys employs an adaptive mechanism where frequently
        accessed data remains in RAM, while less critical information
        gets swapped out to disk storage dynamically. This ensures
        that high-priority tasks receive sufficient resources without
        excessive delays caused by constant disk access. Fascinating
        stuff!<|im_end|>\n<|im_start|>user\nIt does sound fascinating.
         But doesn't relying on virtual memory slow things down
        compared to keeping everything in RAM? What's the balance
        there?<|im_end|>\n<|im_start|>assistant\nGreat question! While
         virtual memory allows for much larger addressable space than
        pure RAM, there is indeed a trade-off in terms of speed.
        Accessing data from disk is significantly slower than
        retrieving it directly from RAM. However, modern systems
        mitigate this issue through intelligent caching strategies.
        For example, frequently used pages are kept in RAM even if
        they originate from virtual memory. Thus, the key lies in
        balancing capacity (via virtual memory) with performance (
        through efficient RAM allocation). It's all about finding the
        sweet spot based on workload characteristics.<|im_end|>\n<|
        im_start|>user\nRefer exclusively to the information in the
        quotation. Rewrite the following explanation about memory
        management in a more readable or differently styled way.",
    "input": "",
    "output": "If an operating system runs low on physical memory, it
        might have to recover unused pages from processes that aren't
        currently utilizing them. Rather than waiting for a process to
         voluntarily relinquish memory, the system proactively takes-
        or \"steals\"-these idle pages and allocates them to other
        processes requiring them.",
    "quote_token_span": [
        [
            52,
            108
        ]
    ],
    "question_token_span": [
        398,
        422
    ]
}
```

**Multi-Span**

```
{
    "instruction": "Could you provide an overview of how major central
        banks manage inflation?<|im_end|>\n<|im_start|>assistant\
        nCertainly. Let us begin with the Federal Reserve. The Federal
         Reserve aims to maintain price stability and maximum
        employment through various monetary policy tools. It primarily
         uses open market operations, adjusting the federal funds rate
        , and setting reserve requirements. By raising or lowering
        interest rates, the Federal Reserve influences borrowing costs
         and overall economic activity. Next, let's discuss the
        European Central Bank. The European Central Bank (ECB) focuses
```

```
            on maintaining price stability within the Eurozone. Its
            primary tool is the adjustment of key interest rates.
            Additionally, the ECB conducts asset purchases and provides
            forward guidance to influence market expectations and
            stabilize inflation over the medium term. Lastly, we will
            cover the Bank of Japan. The Bank of Japan employs a unique
            approach called Quantitative and Qualitative Monetary Easing (
            QQE). This involves expanding the monetary base and purchasing
             a wide range of assets, including government bonds and
            exchange-traded funds. The Bank of Japan also targets a yield
            curve control policy to achieve its inflation objectives.<|
            im_end|>\n<|im_start|>user\nWith only the details given in the
             quotations, list the primary strategies used by the central
            banks to manage inflation.",
    "input": "",
    "output": "The Federal Reserve aims to maintain price stability
            and maximum employment through various monetary policy tools,
            including open market operations, adjusting the federal funds
            rate, and setting reserve requirements.\n The Bank of Japan
            employs Quantitative and Qualitative Monetary Easing (QQE),
            involving the expansion of the monetary base and purchasing a
            wide range of assets, including government bonds and exchange-
            traded funds. It also targets a yield curve control policy to
            achieve its inflation objectives.",
    "quote_token_span": [
        [
            31,
            83
        ],
        [
            156,
            216
        ]
    ],
    "question_token_span": [
        222,
        243
    ]
}
```

**Exclude**

```
{
    "instruction": "I've been reading up on how artificial
            intelligence is being used in healthcare. What are some
            ethical considerations around this topic?<|im_end|>\n<|
            im_start|>assistant\nSure, there are several important ethical
             considerations when it comes to using AI in healthcare. One
            key concern is the reliability and accuracy of AI-driven
            diagnostic tools. These tools can sometimes make errors that
            could lead to misdiagnosis or inappropriate treatments.
            Another significant issue is patient privacy and data security
            . When AI systems process sensitive health data, there's
            always a risk of data breaches or unauthorized access, which
            can compromise patient confidentiality. Lastly, the role of
            human oversight is crucial. Even with advanced AI algorithms,
            having human professionals involved ensures that decisions are
             not solely based on machine outputs, providing an additional
            layer of safety and accountability.<|im_end|>\n<|im_start|>
            user\nIgnoring the selected regions, compile a concise list of
             key points that summarize the main concerns discussed.",
    "input": "",
    "output": "- AI-driven diagnostic tools can sometimes make errors,
             potentially leading to misdiagnosis or inappropriate
            treatments.\n- Human oversight is crucial to ensure decisions
```

```
        are not solely based on machine outputs, providing an
        additional layer of safety and accountability.",
    "quote_token_span": [
        [
            80,
            115
        ]
    ],
    "question_token_span": [
        161,
        179
    ]
}
```

**Info-Combine**

```
{
    "instruction": "I'm curious about sentencing guidelines for drug-
        related offenses. Are they different depending on the
        jurisdiction?<|im_end|>\n<|im_start|>assistant\nYes,
        sentencing guidelines for drug-related offenses vary
        significantly by jurisdiction. For example, in Jurisdiction A,
         the minimum sentence for possession of controlled substances
        is set at 5 years.<|im_end|>\n<|im_start|>user\nThat sounds
        strict. What about other places?<|im_end|>\n<|im_start|>
        assistant\nIn Jurisdiction B, the minimum sentence for
        possession of controlled substances is set at 3 years, which
        is comparatively more lenient than Jurisdiction A.<|im_end|>\n
        <|im_start|>user\nInteresting! Are there any places where the
        penalties are even harsher?<|im_end|>\n<|im_start|>assistant\
        nYes, in Jurisdiction C, the minimum sentence for possession
        of controlled substances is set at 7 years, which is stricter
        than both Jurisdiction A and Jurisdiction B.<|im_end|>\n<|
        im_start|>user\nWhich jurisdiction has stricter or more
        lenient sentencing guidelines for possession of controlled
        substances compared to the jurisdiction(s) I have selected?
        Build your answer solely from the data presented in the
        selected quotations and the related information.",
    "input": "",
    "output": "You have selected Jurisdiction A, where the minimum
        sentence for possession of controlled substances is 5 years.
        Comparing this to the background information, Jurisdiction B
        has a more lenient minimum sentence of 3 years, while
        Jurisdiction C has a stricter minimum sentence of 7 years.",
    "quote_token_span": [
        [
            40,
            62
        ]
    ],
    "question_token_span": [
        177,
        218
    ]
}
```

**Coref**

```
{
    "instruction": "What is the current status of peace negotiations
        between Israel and the Palestinians, and what are the recent
        statements made by Egyptian President Hosni Mubarak and
        Palestinian Leader Yasser Arafat regarding these negotiations
        ?<|im_end|>\n<|im_start|>assistant\nEgypt's President Hosni
```

```
        Mubarak has called on Israeli leaders to stop talking about
        war and to make peace with the Palestinians and Syria. In an
        interview with Israeli Television Monday Mr. Mubarak said
        Israel knows what is necessary for peace, the return of lands
        captured in the 1967 Middle East war. He also acknowledged the
         Palestinian refugees' right of return. In another interview
        Palestinian leader Yasser Arafat said the Israelis and
        Palestinians had made real progress Red Sea resort of Taba,
        but Prime Minister Ehud Barak has suspended contacts with the
        Palestinians until after the February 6th election.
        Correspondent Laurie Cassman has a report. Prime Minister
        Barak's decision not to meet Yasser Arafat this week comes
        after the Palestinian leader's angry speech on Sunday at an
        economic forum in Switzerland. Mr. Arafat accused Israel of
        waging a savage and barbaric war against his people. One day
        earlier Israeli and Palestinian negotiators had announced that
         a week of marathon talks have brought them closer than ever
        to a final agreement. After the angry outburst Mr. Arafat
        reaffirmed his commitment to peace and his willingness to meet
         Mr. Barak. The meeting was expected later this week in
        Stockholm, but Mr. Barak's security advisor told Israeli Radio
         Monday such a meeting now is out of the question. He
        described the Arafat speech as inflammatory. Mr. Barak can not
         risk the controversial meeting ahead of next week's election.
         He still trails far behind the hard line Likud Party leader
        Ariel Sharon in the opinion polls. Laurie Cassman, VOA News,
        Jerusalem.<|im_end|>\n<|im_start|>user\nWhat does the pronoun
        I selected advocate for in this context, particularly
        regarding peace efforts and territorial disputes?",
    "input": "",
    "output": "The pronoun refers to Egypt's President Hosni Mubarak.
        In the given context, Egypt's President Hosni Mubarak calls on
         Israeli leaders to cease discussing war and instead pursue
        peace with the Palestinians and Syria. He emphasizes the
        necessity of returning lands captured during the 1967 Middle
        East war as a key step toward achieving peace. Additionally,
        he acknowledges the Palestinian refugees' right of return.",
    "quote_token_span": [
        [
            114,
            114
        ]
    ],
    "question_token_span": [
        381,
        401
    ]
}
```

# E   Additional evaluations under complex scenarios

*Cross-lingual use (Chinese,* BASE*).*  We instantiate the pipeline in Chinese to test cross-lingual generalization under the same acceptance criteria as in the main data construction.  Results (no language-specific training) are shown in Table 9.

Table 9: Cross-lingual accuracy (%) on a Chinese BASE test set.

| **Variant** | QWEN-2.5-3B-INSTRUCT | LLAMA-3.1-8B-INSTRUCT |
|---|---|---|
| Chinese (BASE) | 93.0 | 87.0 |

*Multi-turn quoting.* We extend the BASE task to a multi-turn variant in which a sequence of questions each quotes a different span from the same dialogue. Results are shown in Table 10, indicating that both single and multi-turn quoting rely on the same span-awareness capability.

Table 10: Accuracy (%) on a multi-turn extension of BASE.

| Variant | QWEN-2.5-3B-INSTRUCT | LLAMA-3.1-8B-INSTRUCT |
|---|---|---|
| Multi-turn quoting | 92.0 | 92.0 |

*Mixed cases.* Our benchmark already contains mixed-capability items that combine MULTI-SPAN with other behaviors. Specifically, EXCLUDE(MULTI-SPAN) accounts for 71.4% of the EXCLUDE set, and INFO-COMBINE(MULTI-SPAN) accounts for 24.4% of the INFO-COMBINE set. Evaluating these subsets separately shows that QUADA remains effective (Table 11).

Table 11: Accuracy (%) on mixed subsets already present in the benchmark.

| Sub-task | QWEN-2.5-3B-INSTRUCT | LLAMA-3.1-8B-INSTRUCT |
|---|---|---|
| EXCLUDE (MULTI-SPAN) | 93.3 | 94.4 |
| INFO-COMBINE (MULTI-SPAN) | 76.2 | 75.0 |

## F   Experiment Details

Both training and inference were conducted on eight H20 GPUs (96 GB each). The inference temperature was set to 1.

**Inference resources**   For Qwen-2.5-14B-Instruct, the baseline model consumed about 32 GB of memory per GPU with a per-sample latency of 0.87 s. Adding QUADA increased memory usage only slightly to 33.7 GB and latency to 1.45 s, indicating minimal overhead.

**Training resources**   Training Qwen-2.5-3B-Instruct for three epochs completed in 1 h 25 min on the same 8×H20 setup, using roughly 28–30 GB of memory per GPU.

All human evaluators held at least a bachelor's degree and relevant research experience.

## G   Ethical considerations and recommended mitigations

**Potential risks.** Parameter-efficient adapters (e.g., LoRA; small bottleneck MLPs) have been shown to carry stealthy Trojans while preserving benign utility [Dong et al., 2023, Liu et al., 2024, Hubinger et al., 2024]. Since QUADA also employs lightweight bottleneck modules, two attack surfaces are relevant: (i) training-time poisoning of adapter weights, and (ii) inference-time manipulation of span inputs (e.g., injected or tampered spans), which could misdirect attention and amplify misinformation.

**Recommended practices.** The following measures are suggested for practitioners and downstream users to reduce risk while preserving utility:

1. *Signed and reproducible releases* Distribute adapter checkpoints with cryptographic hashes (e.g., SHA-256) and, when possible, detached signatures; record the exact backbone commit/version and provide deterministic export instructions so users can verify integrity prior to deployment.

2. *Transparent data and pipeline* Open-sourcing the data-construction pipeline, prompts, and the resulting training/benchmark sets enables third-party auditing for hidden triggers or biases; documenting provenance and filtering criteria further facilitates reproducibility and safety review.

3. *Backdoor-aware training* Incorporate adversarial training in future work to guard against backdoor attacks during training [Hubinger et al., 2024].

4. *Secure span channel* Transmit span annotations over authenticated, encrypted channels; add integrity checks and input sanitization at the boundary where spans enter the model; maintain audit logs to support incident response.

