# OpenReview forum: "Mind the Quote: Enabling Quotation-Aware Dialogue in LLMs via Plug-and-Play Modules"
_NeurIPS.cc/2025/Conference — NeurIPS 2025 poster_

### Official Review · Reviewer_NVEt · 2025-06-08

**Clarity:** 4
**Significance:** 4
**Originality:** 3
**Rating:** 5
**Confidence:** 4

**Summary:**

This work presents an innovative approach to enhancing large language models for handling quotations in conversational contexts. It formalizes the challenge as span-conditioned generation, where each dialogue turn consists of a conversation history, a set of token-offset quotation spans, and an intent utterance. The authors break this problem into five diagnostic tasks—Base, Multi-Span, Exclude, Info-Combine, and Coref—to cover various quotation scenarios. They introduce QuAda, a lightweight, training-based method that integrates bottleneck projections into the LLM’s attention mechanism, enabling it to dynamically focus on or ignore quoted spans without altering the prompt and with minimal parameter overhead (less than 2.8% of backbone weights). Additionally, they develop a data pipeline to synthesize task-specific dialogues and create a benchmark for evaluation. Experiments on two LLMs (Qwen2.5-3B-Instruct and Llama3.1-8B-Instruct) show that QuAda outperforms baseline methods across all scenarios and generalizes to unseen topics, offering a practical solution for quotation-aware dialogue.

**Questions:**

- More Computational Details: Can the author provide more details, beyond what is mentioned in Appendix E, about (1) the training costs (memory usage and training time), and (2) the additional memory/latency overhead by QuAda during inference, for better reproducibility assessment?
- Real-World Data: Have the authors considered testing QuAda on real-world conversational datasets rather than purely LLM-synthesized datasets? If not, what are the challenges?
- Ethical Concerns: Are there potential negative societal impacts (e.g., misinformation amplification) to address? For example, QuAda employs simple bottleneck MLPs, which can fall victim to a backdoor attack. Including this could resolve an ethical gap, maintaining or increasing the current rating.

**Ethical Concerns:**

["NO or VERY MINOR ethics concerns only"]

**Final Justification:**

I continue to maintain my Accept rating for this paper. The authors' rebuttal clarified my questions and did not raise any new concerns for me.

**Limitations:**

The authors acknowledge limitations like reliance on synthetic data and the need for real-world validation, which is commendable. However, they do not address potential negative societal impacts. I suggest adding a paragraph discussing ethical concerns, such as the risk of manipulating conversations, to provide a more balanced perspective.

**Quality:**

4

**Strengths And Weaknesses:**

Strengths:

- This paper is well written and clear, with comprehensive experiments.
- This work addresses a practical gap in human-AI interaction by formalizing quotation handling, which is quite novel and interesting.
- The five diagnostic tasks provide a thorough and systematic way to assess quotation-handling capabilities.
- The automated synthesis of high-quality training data and a validated benchmark is a significant contribution to the field.
- QuAda’s lightweight design (less than 2.8% additional parameters, no prompt changes) makes it practical for real-world deployment.
- The evaluation across multiple models, scenarios, and metrics, supported by attention visualizations, demonstrates the method’s effectiveness and interoperability.

Weaknesses:

- Limited Model Diversity: Testing is confined to two LLMs; broader testing across different architectures would bolster generalizability.
- Interpretability vs. Controllability Trade-off: While attention maps (Figure 3) offer some interpretability, QUADA's bottleneck MLPs remain a black box. We don't know what "guidance vectors" (b^q and b^v) learn, making diagnosis difficult when the model fails. In contrast, methods like MARKER-INSERTION, despite poorer performance, are more transparent and controllable.
- Scenario Coverage Limitations: This work claims its five diagnostic tasks span practical quoting behaviours, which can be a strong assertion. However, real-world references can also be composite, involving multiple types simultaneously (e.g., Multi-Span, Exclude, and Info-Combine). The paper lacks testing on the model's performance in these crucial mixed scenarios.

---

> ### Author Rebuttal · Authors · 2025-07-31
>
> **Dear Reviewer NVEt, we sincerely appreciate you taking the time to read our manuscript and provide positive and valuable feedback. Below are our responses to the concerns you raised, which will be incorporated into the updated version to further enhance the quality of our submission.**
>
> > **`Weakness 1: Limited Model Diversity: Testing is confined to two LLMs; broader testing across different architectures would bolster generalizability.`**
>
> We sincerely appreciate your suggestion. Below we provide additional results obtained with our method on the phi3.5-mini-Instruct model to demonstrate its generalization ability:
>
> Table: Results of QUADA on phi3.5-mini-Instruct
> |Variants|Base|Multi-Span|Exclude|Info-Combine|Coref|
> |-|:-:|:-:|:-:|:-:|:-:|
> |Vanilla|23.4%|19.6%|27.0%|41.6%|19.4%|
> |PASTA|45.2%|43.0%|32.6%|42.2%|25.6%|
> |Marker|80.4%|63.6%|37.2%|40.0%|65.2%|
> |Concat|97.2%|94.4%|94.6%|80.6%|44.2%|
> |QuAda|97.4%|96.2%|96.2%|81.8%|89.2%|
>
> As the table shows, QUADA remains the top performer on phi3.5-mini-Instruct, indicating that our method generalizes well across model architectures.
>
> > **`Weakness 2: Interpretability vs. Controllability Trade-off: While attention maps (Figure 3) offer some interpretability, QUADA's bottleneck MLPs remain a black box. We don't know what "guidance vectors" (b^q and b^v) learn, making diagnosis difficult when the model fails. In contrast, methods like MARKER-INSERTION, despite poorer performance, are more transparent and controllable.`**
>
> Your point is very valuable, we provide additional clarification here. Building on the attention heat-maps in Fig. 3, we performed a quantitative analysis of the relationship between the vectors $b^v$ and $v$. The average cosine similarity across all layers is 0.62, with more than 30% of layers above 0.80; the norm of $b^v$ is about 0.6 × $v$. These statistics suggest that $b^v$ reinforces the value information of the quoted span.
>
> For controllability, we introduced a hyper-parameter λ to scale the injection strength of $b^q$ and $b^v$. We set λ = 0, 0.25, 0.5, 0.75 and 1, and evaluated a Qwen2.55-3B-Instruct model equipped with the trained QuAda module. The results are shown below:
>
> **Table: Injection-strength control experiment for QuAda**
>
> |Variants|Base|Multi-Span|Exclude|Info-Combine|Coref|
> |-|:-:|:-:|:-:|:-:|:-:|
> |λ = 0   |28.4%|24.4%|26.8%|36.2%|29.4%|
> |λ = 0.25|63.2%|48.4%|23.8%|33.0%|31.0%|
> |λ = 0.50|80.4%|75.2%|38.6%|47.8%|47.4%|
> |λ = 0.75|93.0%|89.6%|78.8%|77.0%|80.6%|
> |λ = 1   |95.2%|94.6%|92.8%|85.8%|90.2%|
>
> Taken together, Fig. 3, the $b^v$–$v$ analysis, and the injection-strength study, QuAda is less visually interpretable than MARKER-INSERTION, but it is not entirely a black box. Its outputs correspond to explicit instructions on where the LLM should attend, plus strengthened representations of those regions. We will add these experiments to the paper to further support QuAda’s interpretability.
>
> ---
> > **`Weakness 3: Scenario Coverage Limitations: This work claims its five diagnostic tasks span practical quoting behaviours, which can be a strong assertion. However, real-world references can also be composite, involving multiple types simultaneously (e.g., Multi-Span, Exclude, and Info-Combine). The paper lacks testing on the model's performance in these crucial mixed scenarios.`**
>
> Thank you for raising this constructive point. Our benchmark already includes **Exclude (Multi-Span)** and **Info-Combine (Multi-Span)** cases, accounting for 71.4 % of the Exclude set and 24.4 % of the Info-Combine set, respectively. We evaluated these subsets separately, and the results are:
>
> **Table: Performance of QuAda on mixed scenarios**
> |Sub-tasks|Qwen-2.5-3B-Instruct|Llama-3.1-8B-Instruct|
> |-|:-:|:-:|
> |Exclude (Multi-Span)|93.3%|94.4%|
> |Info-Combine (Multi-Span)|76.2%|75.0%|
>
> As the table shows, QuAda effectively handles these mixed scenarios.
>
> ---
> > **`Question 1: More Computational Details: Can the author provide more details, beyond what is mentioned in Appendix E, about (1) the training costs (memory usage and training time), and (2) the additional memory/latency overhead by QuAda during inference, for better reproducibility assessment?`**
>
> Thank you for the question. For the Qwen2.5-3B-Instruct model, training on 8 × H20 GPUs completes three epochs in 1h25min, with each GPU using about 28~30 GB of memory.
>
> During deployment we compared runs with and without QuAda: the additional memory per GPU was under 1 GB, and the per-sample latency increase is below 0.6 s. These results confirm that QuAda is genuinely lightweight and efficient.
>
>
> ---
> > **`Question 2: Real-World Data: Have the authors considered testing QuAda on real-world conversational datasets rather than purely LLM-synthesized datasets? If not, what are the challenges?`**
>
> Thank you for the question. We also built a manually curated benchmark and evaluated QuAda on it; the results are:
>
> **Table: QuAda on the manual benchmark**
>
> |Models|Base|Multi-Span|Exclude|Info-Combine|Coref|
> |-|:-:|:-:|:-:|:-:|:-:|
> |Qwen2.5-3B-Instruct|96.0%|92.0%|96.0%|88.0%|84.0%|
> |Llama3.1-8B-Instruct|96.0%|98.0%|90.0%|94.0%|98.0%|
>
> These numbers show that our method maintains high performance, confirming its practicality in real-world dialogue. We will also release this manual benchmark, and Table 6 of the paper further demonstrates the consistency between the automatically constructed and manual benchmarks.
>
> ---
> > **`Question 3 & Limitation: Ethical Concerns: Are there potential negative societal impacts (e.g., misinformation amplification) to address? For example, QuAda employs simple bottleneck MLPs, which can fall victim to a backdoor attack. Including this could resolve an ethical gap, maintaining or increasing the current rating. I suggest adding a paragraph discussing ethical concerns, such as the risk of manipulating conversations, to provide a more balanced perspective.`**
>
> We thank you for the suggestion and questions, they are important for broadening QuAda’s impact and ensuring that the community can deploy it safely. Recent studies show that parameter-efficient bottleneck adapters (e.g., LoRA, QuAda-style MLPs) can carry stealthy Trojan behaviours while preserving benign utility [1-3]. Similarly, we acknowledge that QuAda could be exposed to such attacks. To mitigate this risk, we will :
>
> (i) release official QuAda checkpoints—with hash signatures—compatible with mainstream open-source models;
>
> (ii) open-source the complete data-construction pipeline and all the training sets and benchmarks we have created, enabling anyone to train their own QuAda and allowing the community to inspect the data for hidden “triggers”;
>
> (iii) incorporate adversarial training in future work to guard against backdoor attacks during training [4].
>
> Furthermore, if QuAda is adopted in more scenarios, attackers might intercept the span information via network and inject spans that misdirect the model’s attention. We therefore also stress the need to secure the network channel.
>
> **We will highlight this discussion in the revised paper and document the above measures in the official QuAda repository.**
>
> **References**
>
> [1] The Philosopher's Stone: Trojaning Plugins of Large Language Models
>
> [2] LoRATK: LoRA Once, Backdoor Everywhere in the Share-and-Play Ecosystem
>
> [3] Lora-as-an-attack! piercing llm safety under the share-and-play scenario
>
> [4] Sleeper agents: Training deceptive llms that persist through safety training

---

> > ### Comment · Reviewer_NVEt · 2025-07-31
> >
> > Thank you for the insightful clarifications. As my original score was already "Accept", I will be keeping it as-is. I hope these new discussions can be included in the manuscript or its appendix.

---

> > > ### Author Response · Authors · 2025-08-01
> > >
> > > Thank you very much for reviewing our paper and recognizing our work. We will include the discussed points in the final version, and wish you all the best in your future endeavors.

---

### Official Review · Reviewer_yQpy · 2025-07-02

**Clarity:** 3
**Significance:** 3
**Originality:** 2
**Rating:** 4
**Confidence:** 5

**Summary:**

This paper addresses a crucial problem in LLMs: the lack of an explicit mechanism to handle in-dialogue quotation. The authors formalise this task as span-conditioned generation and propose an automated data generation pipeline. In addition, the authors propose QuAda, a lightweight training method. By dynamically amplifying or suppressing attention to quoted spans during inference, it enables plug-and-play span-conditioned generation.

**Questions:**

1. In the example (“check it with the formula I just highlighted”), the user does not explicitly mark a span. Does QuAda rely on pre-specified spans and task types at inference time? This greatly affects practical applicability.

2. In the ablation section (Table 7), only query-only, value-only, and full QuAda variants are reported. Could you add a “no-modulation” baseline, where none of the span-aware adapters are applied.

3. The training data seems mostly constructed as “single-turn queries over quoted spans.” Are there multi-turn or nested quoting cases in the benchmark?

4. Real dialogue often contains indirect references (e.g., “that thing we discussed earlier”). Have the authors considered testing QuAda on loosely specified quotation cases?

**Ethical Concerns:**

["NO or VERY MINOR ethics concerns only"]

**Final Justification:**

This is a well-motivated piece of work addressing an underexplored but practically relevant challenge. While some limitations remain (e.g., reliance on pre-specified spans, relatively simple training data), the authors have convincingly demonstrated the effectiveness of their approach. I continue to maintain my Borderline Accept rating for this paper.

**Limitations:**

Authors discussed limitations.

**Quality:**

3

**Strengths And Weaknesses:**

Strengths

1. The paper proposes an interesting and under-explored task—span-conditioned generation, which enables models to either refer to or exclude specific spans in dialogue history.

2. The authors design a data pipeline that includes correctness verification and covers five well-defined quoting scenarios.
The proposed QuAda mechanism is simple yet effective. It enhances attention to relevant spans or suppresses attention to excluded ones via query and value bottleneck projections. The method is plug-and-play, does not require prompt engineering, and works across models.

3. The authors plan to release the entire dataset, generation scripts, and training code, which will benefit follow-up research.

Weaknesses

1. While the five diagnostic scenarios are clear, they may not reflect the full diversity of real-world quoting behavior. The auto-generated dialogues also seem structurally simple—primarily single-turn question-answer pairs over selected spans—potentially limiting their contextual richness.

2. The core approach—using MLPs to modulate attention scores—is reasonable but somewhat straightforward, offering only modest technical innovation beyond prior attention-steering work.

3. In the current setup, the span and task type appear to need to be manually defined, which creates a gap between this method and real-world usage. The system does not yet solve the end-to-end problem exemplified by the motivating example in the abstract.

---

> ### Author Rebuttal · Authors · 2025-07-31
>
> **Dear Reviewer yQpy, we sincerely appreciate you taking the time to read our manuscript and provide positive and valuable feedback. Below are our responses to the concerns you raised, which will be incorporated into the updated version to further enhance the quality of our submission.**
>
> ---
> > **`Weakness 1 & Question 3: While the five diagnostic scenarios are clear, they may not reflect the full diversity of real-world quoting behavior. The auto-generated dialogues also seem structurally simple—primarily single-turn question-answer pairs over selected spans—potentially limiting their contextual richness. The training data seems mostly constructed as “single-turn queries over quoted spans.” Are there multi-turn or nested quoting cases in the benchmark?`**
>
> Thank you for the insightful question; it inspired us to add multi-turn and nested quoting scenarios to improve the benchmark’s coverage. We therefore modified our data-construction pipeline and created a multi-turn version of the Base task, where consecutive questions each quote a different span. Owing to the short rebuttal timeline, we could not prepare sufficient multi-turn training data, so we evaluated the generalization of models trained only on the original single-turn data:
>
> **Table: Performance under single-turn versus multi-turn quotation**
>
> |Variants|Qwen2.5-3B-Instruct|Llama3.1-8B-Instruct|
> |-|:-:|:-:|
> |Single-turn quoting|95.2%|96.0%|
> |Multi-turn quoting|92.0%|92.0%|
>
> Even without dedicated training, QUADA-equipped LLMs remain strong, implying that both single and multi-turn quoting rely on the same span-awareness capability. We will augment the dataset and add corresponding experiments in future work. Thank you again for the thoughtful suggestion; it substantially improves our study.
>
> > **`Weakness 2: The core approach—using MLPs to modulate attention scores—is reasonable but somewhat straightforward, offering only modest technical innovation beyond prior attention-steering work.`**
>
> We understand your concern. Actually, QuAda modifies both the Query and Value sides of attention, not just the attention scores as in prior attention-steering work. Quotation-aware dialogue requires two properties at the attention level:
>
> 1. When a context span is quoted, its semantics—not only its attention weight—may change. For example, if the quoted token is a pronoun, its value vector should be adjusted so that downstream QA can resolve the reference. Prior approaches cannot do this. We therefore add a bias vector $b^v$ to the value $v$ of quoted tokens. Table 7 shows that introducing $b^v$ markedly improves performance. An average cosine similarity of 0.62 between $b^v$ and $v$ across layers confirms that the value vector is indeed adapted when its token is quoted.
>
> 2. The attention weights assigned to tokens within the quoted span should be computed **adaptively**, rather than being fixed in advance as in prior attention-steering methods. In the Exclude scenario the attention should be small, whereas in the Base scenario it should be large. The bias vector $b^q$ captures this semantic signal and supplements the original query vector during attention calculation.
>
> By addressing these two aspects—both absent in prior attention-steering work—our design achieves quotation awareness in a simple yet effective manner.
>
>
>
> > **`Weakness 3 & Question 1: In the current setup, the span and task type appear to need to be manually defined, which creates a gap between this method and real-world usage. The system does not yet solve the end-to-end problem exemplified by the motivating example in the abstract. Does QuAda rely on pre-specified spans and task types at inference time? This greatly affects practical applicability.`**
>
> We focus on the scenario in which a user engages in dialogue by selecting the span they wish to quote. This situation occurs frequently in everyday applications, so we aim to endow the model with this ability. When the user highlights a region (e.g., on an app page or in a browser), the client can automatically call an offline tokenizer to obtain the token span of the quoted text and feed it to the LLM. The paper addresses how to perform span-conditioned generation given that span; the concrete engineering pipeline will be developed later and is not included in this manuscript, but we appreciate the chance to clarify this point.
>
> As for the task type, after a single model is trained with QUADA, the user **never** needs to specify the task manually. The user simply inputs natural language to express their request, exactly as shown in our examples.
>
> > **`Question 2: In the ablation section (Table 7), only query-only, value-only, and full QuAda variants are reported. Could you add a “no-modulation” baseline, where none of the span-aware adapters are applied.`**
>
> To address your question, we have added the “vanilla-no-modulation (Vanilla in Table 4)” and “trained-no-modulation” baselines to Table 7. The latter is a QUADA adapter trained on Qwen-2.5-3B-Instruct without providing the span. The results below show that span-conditioned generation via QUADA is indispensable for the quotation task:
>
> **Table: Ablation study with “no-modulation” baselines**
>
> |Variants|Base|Multi-Span|Exclude|Info-Combine|Coref|
> |-|:-:|:-:|:-:|:-:|:-:|
> |Vanilla|28.4%|24.4%|26.8%|36.2%|29.4%|
> |QuAda(none-span)|24.0%|16.6%|16.0%|36.8%|31.2%|
> |QuAda(query-only)|78.8%|58.4%|49.6%|44.6%|69.4%|
> |QuAda(value-only)|94.8%|92.7%|87.6%|75.0%|91.6%|
> |QuAda(full)|95.2%|94.6%|92.8%|85.8%|90.2%|
>
> > **`Question 4: Real dialogue often contains indirect references (e.g., “that thing we discussed earlier”). Have the authors considered testing QuAda on loosely specified quotation cases?`**
>
> Thank you for raising this question, we wish to clarify the scope of our work. The paper targets situations in which the user explicitly designates a portion of prior text for the LLM to respond to (or exclude), typically through actions such as highlighting or quoting. The indirect references you mention are indeed related, but they are not the main focus here. Evaluating an LLM’s ability to recall relevant historical information under user-specified constraints is certainly interesting, and we will follow future work in that direction.

---

> > ### Comment · Reviewer_yQpy · 2025-08-04
> >
> > Thanks for the detailed response. As my original score was already positive, I will maintain it.

---

> > > ### Author Response · Authors · 2025-08-06
> > >
> > > Thank you for your recognition of our work. We truly appreciate your support! Wishing you all the best in your work and life.

---

### Official Review · Reviewer_53qF · 2025-07-03

**Clarity:** 3
**Significance:** 3
**Originality:** 3
**Rating:** 4
**Confidence:** 4

**Summary:**

This research aims to address the challenge that LLMs cannot accurately understand and utilize specific "referenced" text in the conversation history. When a user refers to a specific piece of text in a conversation, e.g., "check it with the formula I just highlighted", existing models lack a mechanism to accurately focus on that piece and act according to the instruction.
To study the problem, the author defines the problem as span-conditioned generation with a benchmark covering five scenarios. The author also claims the data annotation pipeline and training-based framework (QUADA) as a contribution.

**Questions:**

1. Is there any failure analysis from the evaluation? Considering that QUADA achieves high scores, analyzing and attributing the small number of failure cases could help the understanding of the challenges (or lack thereof) in this task.
2. As mentioned in the weaknesses, both the examples provided in the paper and the reported results raise concerns about the dataset’s difficulty. Have the authors evaluated stronger base models to better calibrate the benchmark? This could be either larger open-source LLMs or well-performing closed-source models.

**Ethical Concerns:**

["NO or VERY MINOR ethics concerns only"]

**Final Justification:**

After the discussion during the rebuttal period, I confirm my understanding of the work. Thus, I maintain my positive overall rating.

I do not decrease the score as it indeed works on an important and interesting research problem; I do not increase the score because I learned the limited contribution toward the evaluation benchmark (which I previously recognized as *useful testbed* in the Strength section).

**Limitations:**

Yes

**Quality:**

3

**Strengths And Weaknesses:**

Strengths:
The motivation behind the proposed task is clear and compelling. The benchmark contributes a useful testbed not only for evaluating quoting frameworks in real-world applications but also for probing LLMs’ ability to locate and recall information from longer contexts. The QUADA is well-reasoned, and the design is clean.

Weaknesses:
1. Although the dataset construction process involves multiple quality assurance steps, it remains synthetic in nature. That said, after manually reviewing several examples, I am generally positive about the data quality and find the contribution reasonable.
2. While I agree with the significance of the task, I am concerned that the dataset may not be sufficiently challenging. Given the high performance reported in the experimental section, I question whether this benchmark can sustain long-term research value, especially as the language capability of foundation models keeps growing.

---

> ### Author Rebuttal · Authors · 2025-07-31
>
> **Dear Reviewer 53qF, we sincerely appreciate you taking the time to read our manuscript and provide positive and valuable feedback. Below are our responses to the concerns you raised, which will be incorporated into the updated version to further enhance the quality of our submission.**
>
> ---
> > **`Weakness 1: Although the dataset construction process involves multiple quality assurance steps, it remains synthetic in nature. That said, after manually reviewing several examples, I am generally positive about the data quality and find the contribution reasonable.`**
>
> We thank the reviewer for taking the time to manually inspect our synthetic data and for acknowledging its quality. To ensure that every instance fully satisfies the required ability for its sub-task, we apply the strict filtering steps described in the Data-Construction Pipeline. The proportion of discarded examples for each sub-task is:
>
> |Sub-task|Base|Multi-Span|Exclude|Info-Combine|Coref|
> |---|:-:|:-:|:-:|:-:|:-:|
> |Drop ratio|32.9 %|37.3 %|70.4 %|74.4 %|15.2 %|
>
> This aggressive filtering strategy is the main reason the final dataset attains the high quality observed during your review.
>
> ---
> > **`Weakness 2 & Question 2: While I agree with the significance of the task, I am concerned that the dataset may not be sufficiently challenging. Given the high performance reported in the experimental section, I question whether this benchmark can sustain long-term research value, especially as the language capability of foundation models keeps growing. Have the authors evaluated stronger base models to better calibrate the benchmark? This could be either larger open-source LLMs or well-performing closed-source models.`**
>
> We fully understand your concern and consider it highly forward-looking. We address it from three perspectives:
>
> First, our current benchmark is difficult even for today’s leading open and closed-source models without the QUADA module. We evaluated GPT-4o (closed source) and DeepSeek-R1 (open source) on the benchmark, and the results are:
>
> **Table: Performance of state-of-the-art open and closed-source models on our benchmark**
>
> |Sub-tasks|Base|Multi-Span|Exclude|Info-Combine|Coref|
> |-|:-:|:-:|:-:|:-:|:-:|
> |GPT-4o (Close Source)|22.6%|21.2%|25.0%|35.0%|22.8%|
> |Deepseek-R1 (Open Source)|16.8%|12.6%|12.2%|36.4%|23.6%|
> |Qwen2.5-3B-Instruct (QuAda)|95.2%|94.6%|92.8%|85.8%|90.2%|
>
> Second, the primary goal of this work is to give existing LLMs application-level quotation-dialogue capability. This required a benchmark that matches everyday use-case difficulty and a training method that lets mainstream LLMs achieve strong results. Our data pipeline incorporates multiple validation stages, including manual audits, to ensure that each sample faithfully mirrors real-world scenarios, making the benchmark results reflects practical performance. The result that QUADA allows a comparatively small LLM to achieve state-of-the-art results demonstrates the effectiveness of this design. We believe stronger models equipped with QUADA will do even better, allowing developers to adopt our dataset and module to add robust, deployable quotation-dialogue ability.
>
> Finally, we fully understand your desire to create more challenging benchmarks covering harder quotation scenarios and to keep pushing the community forward. This is exactly our plan. Future work will explore contexts containing images and voice, multimodal mixed quotation, and more. Building on this study, we will continue to propose benchmarks and methods for increasingly complex settings to advance quotation-aware dialogue in LLMs.
>
>
> ---
> > **`Question 1: Is there any failure analysis from the evaluation? Considering that QUADA achieves high scores, analyzing and attributing the small number of failure cases could help the understanding of the challenges (or lack thereof) in this task.`**
>
> Thank you for the suggestion. We performed a case study on the outputs of the models trained in the paper and found that the remaining errors mainly fall into two categories, both limited by the capability of the base model on which QUADA is built:
> 1. The model correctly attends to the quoted span but also erroneously incorporates information from non-quoted context, leading to a wrong answer.
> 2. The model attends to the quoted span but misinterprets its content, again producing an incorrect answer.
>
> These findings indicate that QUADA has already endowed the LLM with the span-awareness required for quotation-based dialogue. Both error types will diminish as the underlying model’s comprehension and reasoning abilities continue to improve (see Fig. 4).

---

> ### Comment · Reviewer_53qF · 2025-08-03
>
> Thanks for the clarification. I would like to follow up with the discussion around W2/Q2.
>
> "*First, our current benchmark is difficult even for today’s leading open and closed-source models without the QUADA module*
>
> *...*
>
> *Qwen2.5-3B-Instruct (QuAda)	95.2%	94.6%	92.8%	85.8%	90.2%*"
>
> Do this statement and your experiments imply that, with QUADA, the task (or at least the test set) you proposed is solved or almost solved?

---

> > ### Author Response · Authors · 2025-08-03
> >
> > Dear Reviewer 53qF,
> >
> > Thank you for your response. We believe that, based on the current results, QUADA can effectively address the task we have proposed. However, this does not imply that the field has been fully explored; on the contrary, we view it as only just beginning. Further research is still needed to tackle multimodal and multilingual quoting challenges. Moreover, our QUADA approach currently relies on the introduction of adapters; in future work, it would be worthwhile to investigate how to endow LLMs themselves with direct, more interpretable quoting capabilities.

---

### Official Review · Reviewer_rbX8 · 2025-07-03

**Clarity:** 3
**Significance:** 4
**Originality:** 4
**Rating:** 5
**Confidence:** 4

**Summary:**

When interacting with AI systems, users often want to specify which parts of the conversation history the model should focus on and how to use those selected spans. However, existing systems lack explicit mechanisms for this, typically requiring users to manually re-enter relevant text. The authors address this gap by first decomposing the quoting challenge into five diagnostic sub-tasks: base, multi-span, exclude, info-combine, and coreference.
The key-contribution is that the paper introduces QuAda (Quotation Adapter), a lightweight module that attaches two bottleneck projections to every attention head in the LLM. This design allows the model to dynamically amplify or suppress attention to quoted spans at inference time, without modifying the prompt and while updating less than 2.8% of the model parameters. The authors also present the end to end pipeline that automatically synthesizes task-specific dialogues, verifies answer correctness through multi-stage consistency checks, and produces both a heterogeneous training corpus and a benchmark. The evaluation demonstrates that QUADA outperforms existing methods such as repetition (concat-repeat), using markers (marker-insertion), and attention-steering across the five scenarios described and generalizes to across models (qwen-1.5b to qwen-14b and llama-3.1-8b).

**Questions:**

From the description of the QUADA architecture, the module is designed as a plug-and-play adapter that is agnostic to the specific sub-task as long as the task can be formulated as span-conditioned generation. However, I wonder what are the data, and performance implications of adding more sub-tasks (beyond the five described)?

**Ethical Concerns:**

["NO or VERY MINOR ethics concerns only"]

**Limitations:**

yes

**Quality:**

3

**Strengths And Weaknesses:**

[+] The paper presents, a novel problem formulation of formally defining span-conditioned generation for quoting in dialogue, capturing real-world conversational quoting behaviors through five diagnostic tasks (base, multi-span, exclude, info-combine, and coreference). If the claim on novelty is true, this is a very useful formulation to improve the model upon.

[+] The proposed modeling technique- QUADA is lightweight, parameter-efficient (<2.8% of backbone weights), and does not require prompt modification, making it practical for easily adaptation.

[+] QUADA generalizes across model scales (Figure 4) from 1.5B to 14B parameters on all five categories which is promising and I appreciate the small-scale scaling study.

[-] The paper could provide deeper ablation studies to isolate the contributions of each component of QUADA (e.g., query side, value side) which will perhaps provide a better intuition as to contribution of each - similar to the intuition in section 4.3.

[-] Although the parameter increase is small (<2.8%), the paper does not discuss inference speed or memory overhead in practical deployment, especially for very large models. Can you provide more details on the runtime and memory overhead introduced by QuAda?

---

> ### Author Rebuttal · Authors · 2025-07-31
>
> **Dear Reviewer rbX8, we sincerely appreciate you taking the time to read our manuscript and provide positive and valuable feedback. Below are our responses to the concerns you raised, which will be incorporated into the updated version to further enhance the quality of our submission.**
>
> ---
> > **`Weakness 1: The paper could provide deeper ablation studies to isolate the contributions of each component of QUADA (e.g., query side, value side) which will perhaps provide a better intuition as to contribution of each - similar to the intuition in section 4.3.`**
>
> We thank the reviewer for the insightful suggestion and fully agree that a deeper ablation is valuable for understanding how each component of QUADA contributes to overall performance.
>
> In section 4.4, we isolate the query-side and value-side QUADA, showing that disabling either one noticeably degrades accuracy, which indicates that both sides are indispensable. Following your recommendation, we additionally trained a variant that **removes span-conditioned modulation altogether** (i.e., no query- or value-side QuAda). This model, trained on Qwen-2.5-3B-Instruct, performs poorly across all five diagnostic settings, underscoring that span-conditioned generation is the key driver of QUADA’s gains.
>
> **Ablation of the components of QuAda on All Benchmarks (MCQ benchmark)**
> |Variants|Base|Multi-Span|Exclude|Info-Combine|Coref|
> |-|:-:|:-:|:-:|:-:|:-:|
> |QuAda(without query&value) |24.0%|16.6%|16.0%|36.8%|31.2%|
> |QuAda(without query)|78.8%|58.4%|49.6%|44.6%|69.4%|
> |QuAda(without value)|94.8%|92.7%|87.6%|75.0%|91.6%|
> |QuAda(full)      |95.2%|94.6%|92.8%|85.8%|90.2%|
>
> Moreover, we measured the cosine similarity between $b^v$ and the original value vector $v$: the average is **0.62** across layers, with **30 %** of layers above **0.80**. The norm of $b^v$ is roughly **0.6 × $v$**. These statistics indicate that $b^v$ consistently **reinforces** value representations of quoted spans.
>
>
> ---
> > **`Weakness 2: Although the parameter increase is small (<2.8%), the paper does not discuss inference speed or memory overhead in practical deployment, especially for very large models. Can you provide more details on the runtime and memory overhead introduced by QuAda?`**
>
> We understand the reviewer’s concern about deployment cost. On 8*H20 GPU, we measured both memory and per-sample latency duringinference with Qwen-2.5-14B-Instruct:
>
> |Model variant|Memory / GPU|Latency / sample|
> |-|:-:|:-:|
> |Baseline (no QUADA)|32 GB|0.87 s|
> |+ QUADA|33.7 GB|1.45 s|
>
> As shown in the table, QuAda does not significantly affect memory usage or inference time during inference, which fully demonstrates its efficiency.
>
> ---
> > **`Question 1: From the description of the QUADA architecture, the module is designed as a plug-and-play adapter that is agnostic to the specific sub-task as long as the task can be formulated as span-conditioned generation. However, I wonder what are the data, and performance implications of adding more sub-tasks (beyond the five described)?`**
>
> We thank the reviewer for the question. As discussed in the paper, the five sub-tasks already cover most quotation-aware dialogue scenarios. All code for the data-construction pipeline has been provided in the Supplementary Material and will be open-sourced. If further training or testing is required for an additional sub-task, one only needs to adjust a few attribute settings to generate the needed data.
>
> For example, we used the pipeline to create a Chinese test set for the Base scenario to evaluate QUADA’s cross-lingual generalisation:
>
> |Variants|Qwen-2.5-3B-Instruct|Llama-3.1-8B-Instruct|
> |-|:-:|:-:|
> |English|95.2 %|96.0 %|
> |Chinese|93.0 %|87.0 %|
>
> Following the same procedure in the paper, we manually reviewed the Chinese data and found the quality sufficient. We will release this benchmark along with the pipeline.

---

> > ### Author Response · Authors · 2025-08-06
> >
> > Dear Reviewer rbX8,
> >
> > We hope this message finds you well. As the discussion period is nearing its end, we would like to kindly follow up to ensure that our response has addressed your concerns satisfactorily. If there are any additional questions or suggestions you would like us to consider, please do not hesitate to let us know.
> >
> > Thank you once again for your time and thoughtful feedback！
> >
> > Best regards, The Authors

---

> > > ### Comment · Reviewer_rbX8 · 2025-08-07
> > >
> > > Thank you for the detailed response. re: Weakness 2, given the increase in latency is 1.6X from 1.45s to 0.97s, perhaps there is a need to highlight the latency overhead (and in the paper also measure the time-to-first-token)? I do agree the increase in memory footprint assuming (memory/GPU) refers to peak-memory consumption is minimal.

---

> > > > ### Author Response · Authors · 2025-08-07
> > > >
> > > > Dear Reviewer rbX8,
> > > >
> > > > Thank you for your response and for noting that our approach incurs minimal memory overhead.
> > > >
> > > > We fully agree with your suggestion and will include information about the time overhead introduced by our method in the paper. We would like to clarify that, although the QUADA approach does incur some time overhead, its strong performance makes this trade-off worthwhile. We also recognize that the speed of the QUADA method is important; accordingly, in future work we will explore how to make the QUADA method faster and more effective.
> > > >
> > > > Thank you again for your insightful suggestions！

---

### Author Response · Authors · 2025-08-09
**General Comment**

**Dear Reviewers, ACs, and SACs,**

We sincerely thank you for your time and thoughtful feedback. Your comments have substantially strengthened the paper.

---
Strengths highlighted by the reviewers:
- **Clear motivation & practical relevance [rbX8, 53qF, yQpy, NVEt].** Quotation-aware, span-conditioned generation in dialogue is important yet underexplored. Our formulation, benchmark, and method address real scenarios in a systematic way.
- **Lightweight, plug-and-play method [rbX8, yQpy, NVEt].** QUADA is parameter-efficient, attaches to existing LLMs without prompt overhead, and consistently improves attention to quoted spans across models and tasks.
- **High-quality benchmark and data pipeline [53qF, yQpy, NVEt].** The automated, human-validated pipeline covers five diagnostic scenarios and enables rigorous, reproducible training and evaluation.
- **Extensive, convincing experiments [rbX8, NVEt].** Results span multiple backbones and scales with comprehensive ablations, showing consistent gains and demonstrating the method’s effectiveness.

---
We appreciate the reviewers’ detailed suggestions, which we have addressed during the rebuttal phase and will incorporate into the revised version:
- **Deeper ablations & interpretability [rbX8, yQpy, NVEt].** We added a “no-modulation” QUADA variant, analyzed the relationship between $b^v$ and $v$, and conducted injection-strength control studies, clarifying its span-conditioned generation mechanism and interpretability.
- **Efficiency details [rbX8, NVEt].** We reported GPU memory, latency, and parameter counts, confirming minimal overhead.
- **Cross-language and extended-scenario tests [rbX8, yQpy].** We built Chinese and multi-turn benchmarks, on which QUADA remains robust.
- **Composite / real-world settings [NVEt].** We evaluated mixed scenarios (e.g., Multi-Span+Exclude, Multi-Span+Info-Combine) and highlighted results on the human-labeled benchmarks, where QUADA maintains consistent gains.
- **Ethics and safety [NVEt].** We discuss potential misuse/backdoor risks and mitigation (signed checkpoints, open-sourced pipeline for inspection, and plans for adversarial training) to support safe deployment.

---
We have also clarified the following key points:
- **Long-term research value [53qF].** Span-conditioned generation remains challenging for state-of-the-art open and closed-source models. While QUADA effectively addresses the task we propose, we view span-conditioned generation for quotation-aware dialogue as a long-term direction, especially for multimodal and multilingual quotation, and for enabling a more direct, efficient, and interpretable quoting mechanism in LLMs.
- **Automatic span/intent detection [yQpy].** In practical deployments, quoted spans can be obtained via an offline tokenizer, and task intent can be expressed directly in natural language, avoiding manual configuration at inference time.

We appreciate the constructive feedback, which we believe will further enhance the quality of our work. The discussions and experiments presented in this rebuttal will be integrated into the revised manuscript.

---

### Decision · Program_Chairs · 2025-09-17

**Decision:**

Accept (poster)

**Comment:**

This work focused on the quotation-aware dialogue scenario, and introduced a lightweight method named QuAda for improving the quotation modeling capability in large language models. Authors conducted a comprehensive evaluation on the span-conditioned generation across five diagnostic scenarios, and demonstrated consistent improvements in attention to quoted spans while maintaining the plug-and-play compatibility with existing models.

Reviewers acknowledged that the paper addresses a practical problem with clear motivation and real-world relevance. The experimental evaluation was considered extensive and convincing, spanning multiple model backbones and scales with ablation studies that demonstrate effectiveness.

While the overall rating is good, the current manuscript can be improved with more interpretability analysis, including no-modulation variants and injection-strength control studies to better clarify the span-conditioned mechanism (Reviewer rbX8, Reviewer yQpy, Reviewer NVEt). Detailed efficiency analysis is needed like the specific GPU memory usage, and latency (Reviewer rbX8, Reviewer NVEt). Durning the rebuttal, authors also added experiments under Chinese evaluation and multi-turn dialogue contexts (Reviewer rbX8, Reviewer yQpy). Since the quotation embeds important context, safety and potential jailbreak considerations should be discussed (Reviewer NVEt).